
# GTDI: a gaming integrated drought index implying hazard causing and bearing impacts changing

Xiaowei Zhao[1], Tianzeng Yang[1], Hongbo Zhang[1,2,3*], Tian Lan[1], Chaowei Xue[1], Tongfang Li[1], Zhaoxia Ye[1], Zhifang Yang[1], Yurou Zhang[1]

[1] School of Water and Environment, Chang'an University, Xi'an, 710054, China

[2] Key Laboratory of Subsurface Hydrology and Ecological Effect in Arid Region of Ministry of Education, Chang'an University, Xi'an, 710054, China

[3] Key Laboratory of Eco-hydrology and Water Security in Arid and Semi-arid Regions of the Ministry of Water Resources, Chang'an University, Xi'an 710054, China

Corresponding author: hbzhang@chd.edu.cn

**Abstract:** Developing an effective and reliable integrated drought index is crucial for tracking and identifying droughts. The study employs game theory to create a spatially variable weight drought index (GTDI) by combining two single-type indices: the agricultural drought index (SSMI), which implies drought hazard-bearing conditions, and the meteorological drought index (SPEI), which implies drought hazard-causing conditions. Also, the entropy theory-based drought index (ETDI) is induced to incorporate a spatial comparison to the GTDI to illustrate the rationality of gaming weight integration. Leaf Area Index (LAI) data is employed to confirm the reliability of the GTDI in identifying drought by comparing it with the SPEI, SSMI, and ETDI. Furthermore, an assessment

*Corresponding author: hbzhang@chd.edu.cn



is conducted on the temporal trajectories and spatial evolution of the GTDI-identified drought to
discuss the GTDI's advancedness in monitoring changes in hazard-causing and bearing impacts.
The results showed that the GTDI has a greatly high correlation with single-type drought indices
(SPEI and SSMI), and its gaming weight integration is more logical and trustworthy than the ETDI.
As a result, it outperforms ETDI, SPEI, and SSMI in recognizing drought spatiotemporally, and is
projected to replace single-type drought indices to provide a more accurate picture of actual drought.
Additionally, GTDI exhibits the gaming feature, indicating a distinct benefit in monitoring changes
in hazard-causing and bearing impacts. The case studies show drought events in the Wei River Basin
are dominated by a lack of precipitation. The hazard-causing index SPEI dominates the early stages
of a drought event, whereas the hazard-bearing index SSMI dominates the later stages. This study
surely serves as a helpful reference for the development of integrated drought indices as well as
regional drought mitigation, prevention, and monitoring.
**Keywords:** Integrated drought index; GTDI; drought identification; LAI; Wei River Basin

## 1 Introduction

Drought is one of the most widespread and frequent natural hazards, commonly associated with
inadequate rainfall, a deficit in soil moisture, and reduced stream flow (Berg et al., 2018; Zhang et
al., 2022; AghaKouchak et al., 2023). Due to the combined pressures of climate change and human
activities, the intensity of global drought and the area of arid land have expanded dramatically since
the 21st century (Dai et al., 2013; Huang et al., 2016), severely constraining socio-economic
development and human livelihoods. Moreover, global warming is projected to increase the
frequency and severity of future drought occurrences (Trenberth et al., 2014; Vicente-Serrano et al.,



2020).

China, with its complex terrain and diverse climate types, is one of the countries suffering the

most severe drought-related losses worldwide (Dai et al., 2011; Zhang et al., 2021). Drought is
responsible for more than half of the economic losses caused by climatic hazards in China (Wang et
al., 2023). According to the Ministry of Water Resources of China (MWRC, 2022), the average
annual impacted area of crops and grain loss due to drought was 19.51 million $hm^2$ and 15.8 billion
kg, respectively, from 1950 to 2022. The loss has become increasingly severe, particularly after
2006, resulting in direct economic losses of more than US$ 160 billion in China. For example, the
severe drought event that occurred in southern China from autumn 2009 to spring 2010 deprived
almost 21 million people of drinking water, with direct economic losses of nearly US$3 billion
(Yang et al., 2012). Furthermore, the ongoing drought in China may worsen in the future (Leng et
al., 2015; Wang et al., 2018), with drought occurrences becoming more frequent, intense, and
extended. As a result, scientifically identifying regional drought risks and clarifying regional
drought development and evolution patterns can assist in actively developing drought mitigation
and disaster reduction strategies, assuring the security of food supply and water use.

Drought is currently categorized into four types based on distinct description objects:

meteorological, agricultural, hydrological, and socioeconomic droughts (Wilhite and Glantz, 1985;
Shah and Mishra, 2020). Meteorological drought is characterized by insufficient precipitation,
whereas agricultural drought occurs when soil moisture fails to meet crop development requirements.
Hydrological drought is primarily caused by a lack of surface runoff and groundwater (Xu et al.,
2019; Saha et al., 2023). Socioeconomic drought arises when the aforementioned causes disrupt the
human socioeconomic system, resulting in an imbalance between water supply and demand (Ding

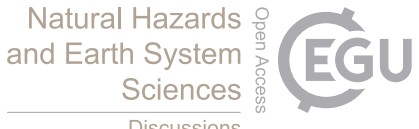

et al., 2021). Despite differing definitions and emphasis, meteorological drought is always regarded
as the root cause of the other three types of drought (Ma et al., 2020). In terms of the driving
mechanism of drought occurrences, meteorological drought indicates the causative attribute of
drought (Zhang et al., 2023), whereas the other three primarily reflect the state of hazard-bearing
entities. Concurrently examining the hazard-causing and hazard-bearing components of drought is
essential for effective estimation and management of drought risk.

Drought is frequently identified using drought indices. The Standardized Precipitation Index

(SPI; Mckee et al., 1993) for meteorological drought, the Standardized Soil Moisture Index (SSMI;
Hao and AghaKouchak, 2013) for agricultural drought, and the Standardized Runoff Index (SRI;
Shukla and Wood, 2008) for hydrological drought are currently the most commonly used drought
indices. These single-type drought indices are primarily used for one-dimensional (type) drought
measurement & evaluation. However, due to the complexity and diversity of drought events, a
single-type drought index is unavoidably insufficient to handle the complete drought development
process (Chang et al., 2016; Wei et al., 2023). As a result, much effort has been expended in
developing comprehensive drought indices, such as the Palmer Drought Severity Index (PDSI;
Palmer, 1965). However, these indices are not very successful at distinguishing between
meteorological and agricultural drought influences and evaluating changes in regional patterns.
Because of this, some works refer to constructing a composite or integrated drought index in two or
more dimensions (Chang et al., 2016; Won et al., 2020; Wei et al., 2023), employing both linear and
nonlinear combination approaches.

The copula function is commonly employed in the nonlinear approach. Won et al. (2020)

proposed a copula-based joint drought index (CJDI) by combining the SPI and the evaporative



demand drought index (EDDI); Wei et al. (2023) used the copula function to connect precipitation,
NDVI, and runoff and then constructed the standardized comprehensive drought index (SCDI),
which has been applied to drought assessment in China's Yangtze River Basin. It should be noted
that copula functions are heavily reliant on the assumption that samples follow a specific probability
density function (Zhang et al., 2019). However, due to the complicated interactions between the
atmosphere, vegetation, soil, and groundwater, the drought does not generally meet it. If the copula
function is used to estimate drought quantiles, significant biases may be introduced, affecting the
reliability of the copula-based integrated drought indices (Huang et al., 2015).
A comprehensive drought index can also be generated by linearly mixing single-type drought
indices, such as the entropy weight method (Huang et al., 2015) and the principal component
analysis method (Liu et al., 2019). In the relevant research, it is highly emphasized that the weighting
of different types of drought indices is critical since it has a significant impact on the reliability of
drought monitoring results (Liu et al., 2019; Wei et al., 2023). Furthermore, it has been revealed that
the impacts of different factors on drought, such as hazard-causing and hazard-bearing, are changing
spatially and game-playing, necessitating the development of effective linear combination methods
for measuring their spatial heterogeneity in contribution to drought. Therefore, game theory is
suggested for the integration of drought indices because it can comprehensively consider the
opinions of each party to achieve a distribution pattern that satisfies each participant (Lai et al., 2015;
Jato-Espino and Ruiz-Puente, 2021), and has been widely applied in water resources management
(Madani, 2010; Khorshidi et al., 2019; Batabyal and Beladi, 2021).
This study proposes a game theory-based drought index (GTDI), which integrates the
meteorological drought index SPEI, implying hazard-causing impact, and the agricultural drought



index SSMI, implying hazard-bearing impact, through the game theory method. The structure of
this study is as follows: Section 2 introduces the research topic and data source. Section 3 describes
the SPEI, SSMI, GTDI, and ETDI (entropy theory-based drought index) calculation procedures, as
well as the verification and analysis methodologies. Section 4 investigates the evolutionary features
of GTDI, examines its rationality of integrated weight in comparison to ETDI, and validates its
usefulness in identifying drought occurrences using Leaf Area Index (LAI) data. Furthermore, the
impact of hazard-causing and bearing indices on GTDI's spatiotemporal evolution is explored
through the synergistic analysis of GTDI, SPEI, and SSMI. Finally, Section 5 highlights the study's
significant findings.

## 2 Study area and data

### 2.1 Study area

The Wei River is the largest tributary of the Yellow River, with a drainage area of 134,800 km$^2$ (Fig.
1). It rises to the north of Niaoshu Mountain in Gansu Province, about 33.5°–37.5°N latitude and
103.5°–110.5°E longitude, and runs primarily through Shaanxi, Gansu, and Ningxia provinces. The
Wei River Basin (WRB) is high in the west and low in the east, with a geographical elevation ranging
from 322 to 3777 meters. The WRB has a continental monsoon climate with large seasonal
fluctuations, with average annual temperatures and precipitation ranging from 7.8 to 13.5°C and
500 to 800 mm, respectively (Zhang et al., 2022). Precipitation in the WRB accounts for over 60%
of the total annual amount, and its spatial distribution shows a steady decrease from southeast to
northwest. Furthermore, evaporation is significant in the WRB, with annual water surface
evaporation ranging from 660 to 1600 mm. As a result of its specific climate characteristics, the
WRB is a typical place for drought research.

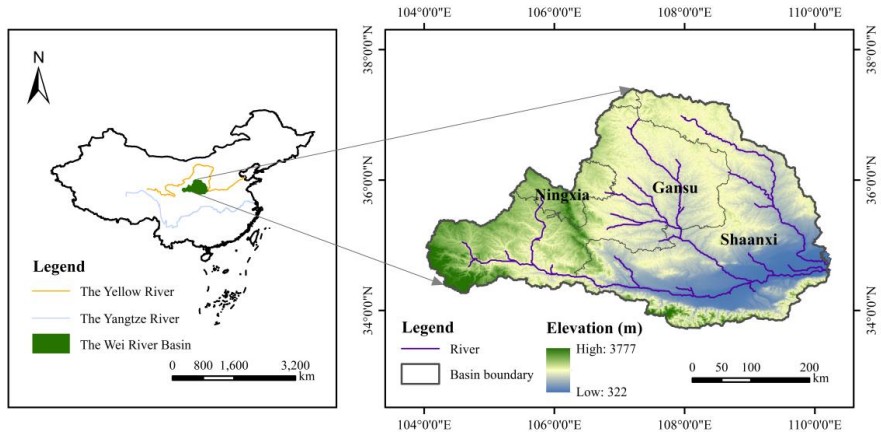


**Figure 1.** A map of the Wei River Basin.
**2.2 Data source**
The data used in this study comprises: (1) DEM data with a grid size of 30 m; (2) monthly
precipitation and temperature dataset from 1950 to 2020 with a grid size of 1 km; (3)
GLDAS_NOAH025_3H_2.0 and GLDAS_NOAH025_3H_2.1's soil moisture dataset for 0 to 10
cm of soil surface layer, with a spatial resolution of 0.25° and data period from 1950 to 2020; (4)
GLOBMAP leaf area index dataset (Version 3) with a period of 1981 to 2019 and a spatial resolution
of 0.08°. Additionally, in order to facilitate calculation and analysis, precipitation, air temperature,
soil moisture, and leaf area index (LAI) data were all resampled to the same spatial resolution of
0.125° in this study. The data source is shown in Table 1.
**Table 1.** Data source.

| Name | Source |
| --- | --- |
| DEM data | http://www.ncdc.ac.cn/ |
| Precipitation dataset | http://www.geodata.cn/ |
| Temperature dataset | http://www.geodata.cn/ |
| Soil moisture dataset | https://disc.gsfc.nasa.gov/datasets/ |





| LAI dataset | https://www.resdc.cn/ |

## 3 Methodology

### 3.1 Calculation of single-type drought indices

#### 3.1.1 SPEI

The Standardized Precipitation Evapotranspiration Index (SPEI) was first introduced by Vicente Serrano et al. in 2010. As a meteorological drought index, SPEI primarily characterizes the hazard-causing attribute of drought (Zhang et al., 2023). On the basis of the Standardized Precipitation Index (SPI), SPEI takes potential evapotranspiration (PET) into account and demonstrates superior effectiveness and applicability (Labudová et al., 2017; Li et al., 2020; Tan et al., 2023). The Thornthwaite method, which can better reflect the potential surface evapotranspiration, is employed to calculate PET in this paper. As is well known, drought indices on different time scales can reflect the dry and wet conditions of the study area during different periods. In this study, we calculated the SPEI series over a three-month timescale. The detailed calculation procedure for SPEI can be found in Vicente Serrano et al. (2010).

#### 3.1.2 SSMI

Drought can have a direct impact on the growth state of hazard-bearing bodies such as crops (Zhang et al., 2023), making agricultural drought hazard-bearing. The Standardized Soil Moisture Index (SSMI) is one of the most effective indices for predicting agricultural drought (Hao et al., 2013), and its calculation method is comparable to that of the SPI (Xu et al., 2021; You et al., 2022). Meanwhile, it was revealed that the log-logistic probability distribution function with three parameters was better suited to soil moisture data sequences than the original gamma probability





distribution function (Oertel et al., 2018). As a result, in this study, we employed the calculation
method proposed by Oertel et al. for the agricultural drought index SSMI, with a three-month time
scale, just like the SPEI.
**3.2 Construction of integrated drought indices**
In this study, two integrated drought indices, GTDI and ETDI, are built utilizing game theory and
the entropy weight method for index weight allocation, respectively, and both combine SPEI and
SSMI. ETDI serves as a comparison to GTDI, and Huang et al. (2015) provide the computation
process for it.
As a subset of optimality modeling, game theory (GT) investigates the interacting outcomes of
resource conflicts and cooperation between two or more entities (Lai et al., 2015). It attempts an
optimal allocation approach that maximizes the interests of each participant through mathematical
analysis (Jato-Espino and Ruiz-Puente, 2021). Currently, GT has been widely applied in the fields
of hydrology and water resources, such as water price equilibrium (Batabyal and Beladi, 2021),
reservoir scheduling policy (Khorshidi et al., 2019), and subjective/objective weighting issues (Liu
et al., 2020). In this study, the hazard-causing index (SPEI) and the hazard-bearing index (SSMI)
are regarded as two opponents in the game. Through confrontation, the GT technique gets the ideal
weight allocation for both sides and then uses this to produce the integrated drought index (GTDI)
at each grid point. The following are the methods for creating GTDI using game theory:
**Step 1:** A possible weight set is combined by SPEI and SSMI in the form of an arbitrary linear
combination as follows:

$$V = \alpha_{spei} V_{spei}^T + \alpha_{ssmi} V_{ssmi}^T, (\alpha_{spei}, \alpha_{ssmi} > 0) \tag{1}$$





Where $V$ is a possible combined vector, $V_{spei}$ & $V_{ssmi}$ are the weight vectors of SPEI and SSMI, and
$\alpha_{spei}$ & $\alpha_{ssmi}$ are the weight coefficients.

**Step 2:** Minimize the deviation between $V$ and $V_k$ using the following formula:

$$\text{Min} \left\| V - V_k \right\|_2 , (k = spei, ssmi) \tag{2}$$

**Step 3:** According to the differentiation property of the matrix, transform formula (2) into a

first-order system of linear equations:

$$\begin{bmatrix} V_{spei}V_{spei}^T & V_{spei}V_{ssmi}^T \\ V_{ssmi}V_{spei}^T & V_{ssmi}V_{ssmi}^T \end{bmatrix} \begin{bmatrix} \alpha_{spei} \\ \alpha_{ssmi} \end{bmatrix} = \begin{bmatrix} V_{spei}V_{spei}^T \\ V_{ssmi}V_{ssmi}^T \end{bmatrix} \tag{3}$$

**Step 4:** Solve the weight coefficients $\alpha_{spei}$ and $\alpha_{ssmi}$ in equation (3) and normalize them.

$$\begin{cases} \alpha_{spei}^* = \alpha_{spei} / \left( \alpha_{spei} + \alpha_{ssmi} \right) \\ \alpha_{ssmi}^* = \alpha_{ssmi} / \left( \alpha_{spei} + \alpha_{ssmi} \right) \end{cases} \tag{4}$$

**Step 5:** Calculate GTDI:

$$V_{gtdi} = \alpha_{spei}^* V_{spei}^T + \alpha_{ssmi}^* V_{ssmi}^T \tag{5}$$

Where $V_{gtdi}$ is the combined vector of GTDI, $\alpha_{spei}^*$ and $\alpha_{ssmi}^*$ are the normalized weight coefficients of
SPEI and SSMI, respectively.

## 3.3 Classification criteria for drought

**Table 2.** Drought classification criteria for the SPEI, SSMI, GTDI and ETDI.

| Grade | Classification | Values |
|---|---|---|
| 1 | No drought | -0.5< Index |
| 2 | Mild drought | -1.0< Index ≤ -0.5 |
| 3 | Moderate drought | -1.5< Index ≤ -1.5 |
| 4 | Severe drought | -2.0< Index ≤ -1.5 |
| 5 | Extreme drought | Index ≤ -2.0 |

The calculating approach of SSMI in this study is comparable to that of SPEI, while GTDI and
ETDI are built on SSMI and SPEI. As a result, as indicated in Table 2, the SSMI, GTDI, and ETDI



use the same grading criteria as the SPEI.

## 3.4 Reliability verification

### 3.4.1 Evaluation of correlation

A correlation analysis of the integrated drought index with two single-type drought indices is
necessary to assess the consistency of indicators before and after coupling. Thus, the Pearson's
correlation coefficients (PCC) between GTDI/ETDI with SPEI and SSMI are calculated for each
grid (Eq. 6), and their correlation in different locations is explored. Table 3 shows the correlation
levels and corresponding absolute value range of PCC.

$$PCC_{x,y} = \frac{\sum_{i=1}^{n}\left(x_i - \bar{x}\right)\left(y_i - \bar{y}\right)}{\sqrt{\sum_{i=1}^{n}\left(x_i - \bar{x}\right)^2 \sum_{i=1}^{n}\left(y_i - \bar{y}\right)^2}} \qquad (6)$$

Where $n$ denotes the sample size; $x_i$ and $y_i$ are data samples of $x$ and $y$, respectively; $\bar{x}$ and $\bar{y}$ are
arithmetic average of $x$ and $y$, respectively.
**Table 3.** The absolute value range of PCC and correlation levels.

| Correlation levels | Absolute values of PCC |
| --- | --- |
| Greatly low or none | [0, 0.2] |
| Low | (0.2, 0.4] |
| Moderate | (0.4, 0.6] |
| High | (0.6, 0.8] |
| Greatly high | (0.8, 1.0] |

### 3.4.2 Efficacy verification in identifying drought

Because surface vegetation is highly sensitive to soil moisture (Li et al., 2022), drought usually leads
to a decrease in vegetation Leaf Area Index (LAI; Fang et al., 2019; Bock et al., 2023). In light of
this, LAI data are used to evaluate the drought recognition capabilities of various indexes to further
validate their dependability. The leaf area index dataset used is the GLOBMAP leaf area index


212 product (https://www.resdc.cn/).

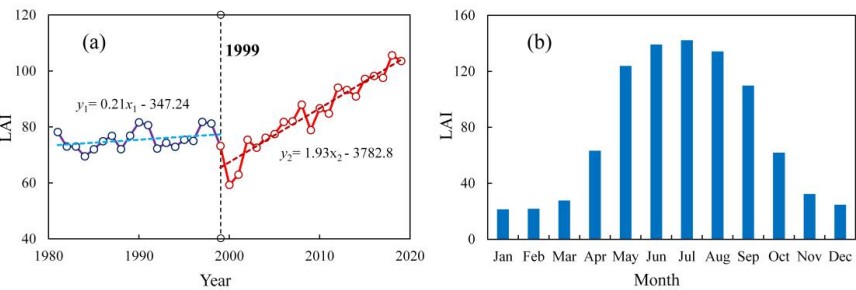


**Figure 2.** The plot graphs of the Leaf Area Index (LAI) in the Wei River Basin with an interannual

trend spanning from 1981 to 2019 (a) and the average monthly allocation from 1981 to 1999 (b).

216   Significant disparities in LAI trends can be identified in the WRB around 1999, as illustrated

217 in Fig. 2(a). Prior to 1999, the average annual growth rate of LAI was only 0.21/a, but it skyrocketed

218 to 1.93/a after 1999, owing mostly to "Grain for Green" (Li et al., 2019; Tian et al., 2022). In order

219 to mitigate the potential inaccuracy resulting from the regional LAI trend change, we selected the

220 validation years of 1981 to 1999, during which the growth trend was relatively weak. Also, LAI in

221 the WRB rises significantly from March to August, falls fast from September to November, and then

222 remains low from December to January of the following year (Fig. 2b). It can be discovered that

223 LAI's trend change in autumn and winter is the result of vegetation's natural growth cycle, resulting

224 in a reduced sensitivity of LAI to soil moisture and further failing to identify drought. As a result,

225 the autumn and winter months (September to January) should also be excluded from the validation

226 period.

227   In summary, LAI raster data from March to August (spring and summer) of the period from

228 1981 to 1999 were selected to verify the drought identification efficacy of drought indices.

229 Meanwhile, the image from the mid-month of each month is regarded as the representative data of



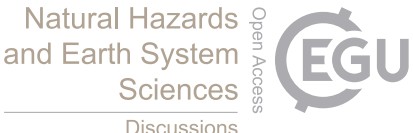

the month. If the occurrence of drought has been discovered, it can be determined by comparing the
mean drought index values during arid months with non-arid months. The specific process is as
follows:

$$\begin{cases} M_{d,i} = \dfrac{\sum_{j=1}^{m} I_{i,j}}{m} \\ M_{n,i} = \dfrac{\sum_{l=1}^{n} I_{i,l}}{n} \end{cases} \tag{7}$$

$$R_i = \begin{cases} 1, M_{d,i} < M_{n,i} \\ 0, M_{d,i} \geq M_{n,i} \end{cases} \tag{8}$$

Where $M_{d,i}$ and $M_{n,i}$ represent the average values of the drought index in the $i$-th grid during arid
and non-arid months, respectively; $m$ and $n$ are the number of arid and non-arid months, respectively;
$I_{i,j}$ and $I_{i,l}$ represent the drought index value of the $i$-th grid during the $j$-th arid month and the $l$-th
non-arid month, respectively; $R_i$ represents the drought recognition performance of the drought
index in the $i$-th grid, with a value of 1 indicating fine and 0 indicating poor.

## 3.5 Analysis methods for drought characteristics

### 3.5.1 Mann-Kendall test

The Mann-Kendall (M-K) test is a non-parametric statistical test method with a simple
computational process. It has been extensively utilized for the analysis of hydrological and
meteorological sequences (Zhang et al., 2021; Agbo et al., 2023). In this study, the M-K test method
is used to perform trend testing on the drought index sequences, and the calculation principle can
be referred to Cai et al. (2022).

### 3.5.2 Drought identification

Drought is often identified by two factors: the drought index threshold and the drought area


threshold. In this study, we used -1 as the drought index threshold, which is compatible with current
research (Deng et al., 2021; Feng et al., 2023), and 1.6% as the area threshold (Wang et al., 2011).
Furthermore, a spatiotemporal continuity technique is used to detect drought occurrences, with
specific procedures available in Deng et al. (2021).
**3.5.3 Spatiotemporal characteristics of drought**
The spatiotemporal characteristics of drought mostly manifest in variables such as drought intensity,
drought area, drought duration, and drought centroid (Wen et al., 2020). Based on the current
research methods for studying the spatiotemporal characteristics of drought, we divided the
variables representing drought characteristics into two scales: grid point and monthly, in order to
systematically analyze and describe the drought characteristics of the WRB.
(1) Grid point's drought characteristic variable
The drought intensity $S_i$ of the grid point is calculated by:

$$S_i = S_0 - I_i \tag{9}$$

Where $I_i$ is the value of the $i$-th drought grid point; $S_0$ is the threshold of the drought index.
(2) Monthly drought characteristic variables
The monthly drought characteristic variables consist of the monthly drought intensity $S_{am}$, the
monthly drought area $A_{am}$, and the monthly drought centroid ($X_{am}$, $Y_{am}$), as shown in Table 4.
**Table 4.** Monthly drought characteristic variables.

| Variables | Formula | Notes | Number |
|---|---|---|---|
| Monthly drought intensity $S_{am}$ | $S_{am} = \dfrac{1}{k} \sum_{i=1}^{k} S_i$ | Where $k$ is the number of drought grids; $S_i$ is the intensity value of the $i$-th drought grid. | (10) |
| Monthly drought area $A_{am}$/$10^4$km$^2$ | $A_{am} = kA$ | Where $A$ is the spatial range of a single grid, and its unit is $10^4$ km$^2$. | (11) |





| Monthly drought centroid $(X_{am}, Y_{am})$ | $\begin{cases} X_{am} = \sum_{i=1}^{k} S_i x_i \Big/ \sum_{i=1}^{k} S_i \\ Y_{am} = \sum_{i=1}^{k} S_i y_i \Big/ \sum_{i=1}^{k} S_i \end{cases}$ | Where $S_i$ is the drought intensity value of the $i$-th drought grid, and $x_i$ and $y_i$ are the longitude and latitude coordinates of the $i$-th drought grid, respectively. | (12) |

## 4 Results and Discussion

### 4.1 Evolutionary characteristics of integrated drought index GTDI

Using the game theory method, the monthly GTDI of the WRB was calculated based on SPEI and SSMI. Meanwhile, considering the WRB's seasonal characteristics, GTDI sequences from May, August, November, and February of the next year were chosen to represent the drought conditions of spring, summer, autumn, and winter, respectively.

Fig. 3(a) demonstrates the temporal evolution characteristics of the monthly GTDI in the WRB from 1950 to 2020. Therein, the linear tendency rate of GTDI is -0.024/10a, illustrating that the drought in the WRB is aggravating, which is also mentioned in Wang et al. (2020). Particularly since the 1990s, the frequency of moderate and severe drought months and their average drought intensity have increased by 5.1% (from 34.1% to 39.2%) and 0.043 (from 0.242 to 0.285), respectively. In terms of seasonal change, drought in the WRB showed an increasing trend in spring, summer, and autumn (Fig. 3b-d). In the eastern half of the WRB, the significantly aggravated area of spring drought accounts for 29.98% of the overall basin, while most places in summer and autumn show a non-significant aggravation in drought severity. Winter is an exception, as most areas experience a reduction in drought, especially in the eastern and northern regions of the WRB (Fig. 3e).


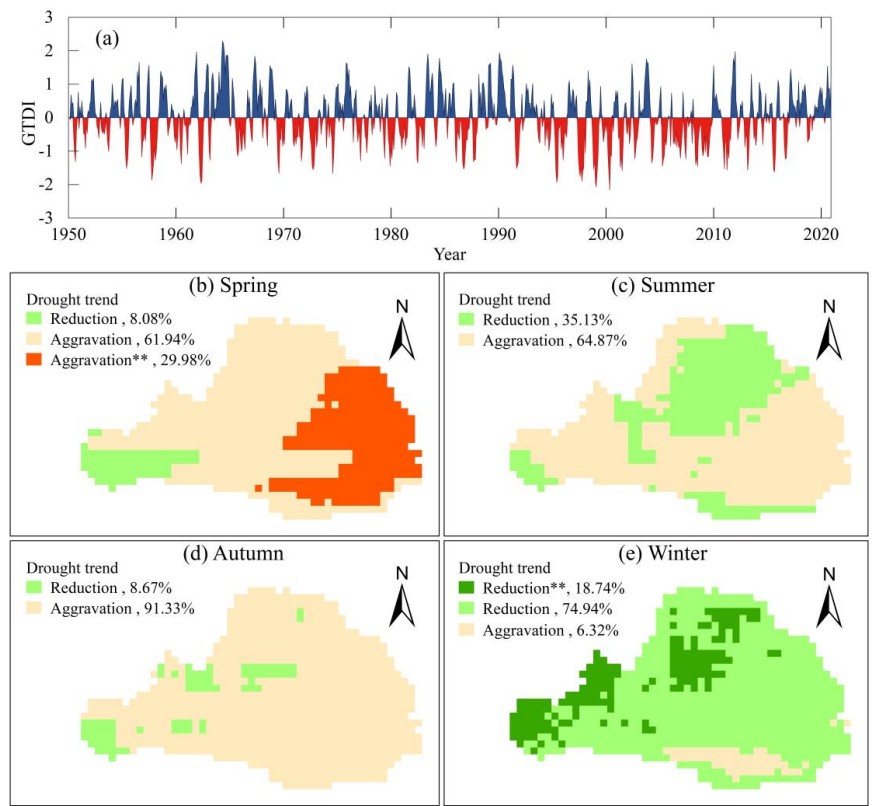

**Figure 3.** Temporal evolution characteristics of integrated drought in the Wei River Basin from 1950

to 2020 (a), and spatial distribution of drought trends in different seasons (b-e). The symbol "**"

donates the change is significant, and the percentage means the area proportion of different trend

types.

## 4.2 Reliability verification of the GTDI

### 4.2.1 The evaluation of correlation

Table 5 illustrates the grid proportions of different correlation levels between the integrated drought

indices (GTDI and ETDI) and the single-type drought indices (SPEI and SSMI), whereas Fig. 5

depicts the spatial distribution of their correlation coefficients in different seasons.


**Table 5.** Grid proportions of integrated drought indices (GTDI, ETDI) and single-type drought
indices (SPEI, SSMI) at different correlation levels.

| Correlation levels | GTDI vs. SPEI | | | | GTDI vs. SSMI | | | |
|---|---|---|---|---|---|---|---|---|
| | Spring | Summer | Autumn | Winter | Spring | Summer | Autumn | Winter |
| Greatly high | 100% | 100% | 100% | 100% | 100% | 100% | 100% | 54.8% |
| High | 0 | 0 | 0 | 0 | 0 | 0 | 0 | 45.2% |

| Correlation levels | ETDI vs. SPEI | | | | ETDI vs. SSMI | | | |
|---|---|---|---|---|---|---|---|---|
| | Spring | Summer | Autumn | Winter | Spring | Summer | Autumn | Winter |
| Greatly high | 83.6% | 89.5% | 88.4% | 66.2% | 89.7% | 95.6% | 98.2% | 68.3% |
| High | 16.4% | 10.5% | 11.6% | 33.3% | 10.3% | 4.4% | 1.8% | 25.8% |
| Moderate | 0 | 0 | 0 | 0.5% | 0 | 0 | 0 | 5.4% |
| Low | 0 | 0 | 0 | 0 | 0 | 0 | 0 | 0.5% |

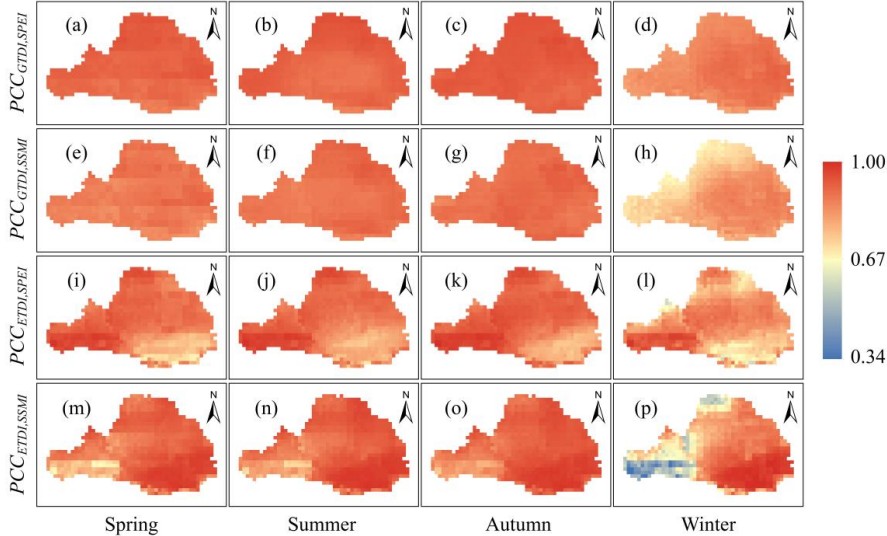

Spring          Summer          Autumn          Winter

**Figure 4.** Spatial distribution of correlation coefficients in different seasons. The color bar on the
right denotes the correlation coefficients.
As shown in Table 5 and Fig. 4, the correlation between GTDI and SPEI or SSMI in the entire
WRB is quite significant, and the correlation coefficients (PCC) are close to 1 in spring, summer,
and autumn, but slightly worse in winter (Fig. 4a-h). The correlation coefficients in the western and
northern areas of the WRB are lower in winter (Fig. 4d, h, l, p), but the minimal correlation





coefficients between GTDI and SPEI or SSMI are still above 0.83 and 0.67, respectively (Fig. 4d,
h). It is worth noting that GTDI and SPEI have a greatly high correlation across the WRB over all
four seasons, whereas 45.2% of locations only have a good correlation between GTDI and SSMI in
winter (Table 5). As a result, the correlation between GTDI and SPEI is stronger than that of SSMI,
especially during the winter season.
The graph also shows that the integrated drought index (ETDI) demonstrates spatially opposite
correlations with SPEI and SSMI. For instance, in the southeastern area of the Wei River Basin,
there is the worst association between ETDI and SPEI, but the correlation between ETDI and SSMI
is the strongest (Fig. 4i-p). Similar to GTDI, the correlation between ETDI and SPEI or SSMI is
slightly higher in spring, summer, and autumn than in winter. However, as compared to GTDI, the
geographical variability of the correlation coefficients between ETDI and SPEI or SSMI is more
pronounced in the same season (Fig. 4). As seen in winter (Fig. 4p), the highest correlation
coefficient between ETDI and SSMI is approximately 1, while the lowest value is around 0.34. In
terms of grid proportions at various levels of correlation, the correlations between ETDI and SPEI
or SSMI do not achieve a greatly high level in certain regions over the four seasons (Table 5),
resulting in their performance falling short compared to GTDI.
Overall, GTDI exhibits superior performance to ETDI, particularly in terms of the homogeneity
of the spatial distribution of correlation coefficients, indicating that the integrated drought index
GTDI constructed in this study has more reliable consistency with single-type drought indices (SPEI
and SSMI).
**4.2.2 Comparison of the integrated weight of GTDI and ETDI**
To contrast the weight distribution of SPEI and SSMI in creating the integrated drought indices
GTDI and ETDI, the spatial distribution of their weight ratios (SPEI/SSMI) in the WRB is plotted,
as shown in Fig. 5.

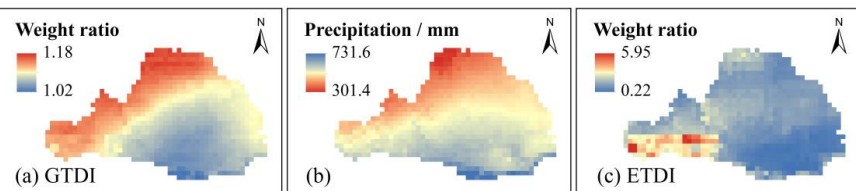

**Figure 5.** Comparison of the integrated weights of GTDI and ETDI. Subfigures (a) and (c)
demonstrate the spatial distribution of weight ratio (SPEI/SSMI) in the construction process of
GTDI and ETDI, respectively, and (b) is a spatial distribution map of the average annual
precipitation in the Wei River Basin.

The GTDI, a comprehensive drought index constructed using the game theory method, exhibits

a spatial distribution of the weight ratio (SPEI/SSMI) that gradually decreases from northwest to
southeast (Fig. 5a). Furthermore, the weight ratio in GTDI ranges from 1.02 to 1.18, showing a
substantially balanced weight allocation between the hazard-causing index (SPEI) and the hazard-
bearing index (SSMI). Meanwhile, the weight ratio of SPEI to SSMI is closer to 1 in places with
greater precipitation (Fig. 5a-b). It is noteworthy that the change in weight ratio (SPEI/SSMI) in
GTDI closely resembles the spatial distribution pattern of the average annual precipitation in the
WRB, as evidenced by a correlation coefficient of -0.88, indicating a significant negative
relationship.

The entropy theory-based drought index (ETDI), on the other hand, does not show a distinct

spatial distribution pattern for the weight ratio of SPEI to SSMI. Its weight ratio fluctuates greatly
between locations, ranging from 0.22 to 5.95 (Fig. 5c), implying that entropy theory fails to establish
a consistently stable allocation of weights in the integrated drought index ETDI development





process. Furthermore, the weight ratio (SPEI/SSMI) in ETDI has a low relationship with annual
average precipitation, as evidenced by a correlation coefficient of only -0.04.
As a consequence of comparing GTDI and ETDI, it is discovered that the game theory
approach gives an integrated weight geographic distribution compatible with the precipitation-
dominated natural drought pattern, which is essentially congruent with the drought generation
mechanism in this basin. As a result, it is thought that the weighting of SPEI and SSMI in GTDI is
more reasonable and reliable.
**4.2.3 The efficacy verification in identifying drought**
To further investigate the reliability of the integrated drought index GTDI, the Leaf Area Index (LAI)
data is used to assess its efficacy in identifying drought, and the drought recognition performance
of the GTDI is evaluated by Eq. 8 and presented in Fig. 6. To compare, Fig. 7 depicts the spatial
distribution of efficacy in recognizing drought using the ETDI, SPEI, and SSMI, and Table 6
provides a statistical list exhibiting the efficacy ratios of four drought indices in different validation
months.
**Table 6.** The efficacy ratios of four drought indices in different validation months

| Drought indices | March | April | May | June | July | August |
|---|---|---|---|---|---|---|
| GTDI | 78.6% | 84.1% | 90.4% | 71.8% | 87.5% | 76.3% |
| ETDI | 48.4% | 49.6% | 50.7% | 50.5% | 49.2% | 48.6% |
| SPEI | 50.1% | 49.5% | 50.6% | 49.4% | 48.4% | 48.8% |
| SSMI | 49.1% | 50.4% | 52.8% | 49.9% | 49.5% | 48.9% |

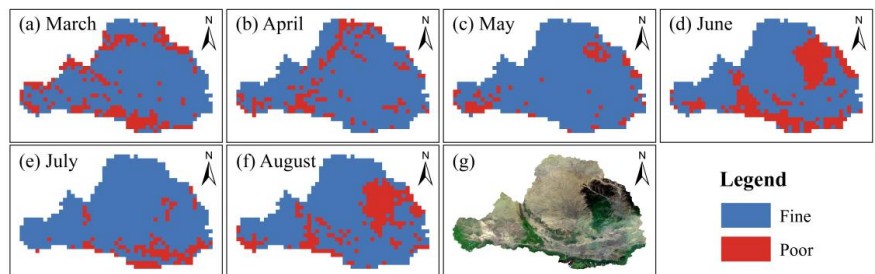

**Figure 6.** The spatial distribution of GTDI's efficacy in identifying drought in the Wei River Basin.

Subfigures (a)-(f) depict the findings from March to August, and (g) displays a satellite image of the

Wei River Basin.

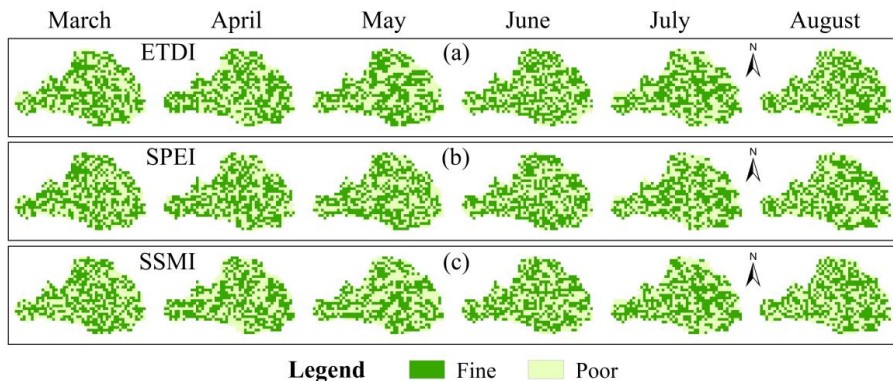

**Figure 7.** The spatial distribution of efficacy in identifying drought of the ETDI, SPEI and SSMI.

During the validation period from March to August, GTDI performs well in recognizing

drought (Fig. 6), particularly in May, when it captures 90.28% of the drought in the WRB (Table 6).

GTDI, on the other hand, performs relatively badly in June (Fig. 6d) and August (Fig. 6f), only with

71.8% and 76.3% of effective recognition grid points, respectively (Table 6). In conjunction with

Fig. 6(g), it is discovered that the grid points with poor performance in June and August are

concentrated in the forest area, which is the dark green area in the WRB's northeast hinterland. As

is widely known, forests have more access to deeper soil moisture than farming land and grassland

(Xu et al., 2018; Bai et al., 2023), resulting in forests having higher drought tolerance than other



terrestrial vegetation types (Jiang et al., 2020; Chen et al., 2022). However, the soil moisture data
used in this study is only 0 to 10cm of soil surface layer, which could explain why GTDI's drought
diagnosis ability in the forest region is skewed. Even with the defect in forest regions, GTDI has
exhibited strong drought monitoring capabilities in the WRB, and can effectively capture the
occurrence of drought.
In contrast to GTDI, the effectiveness of drought detection by ETDI, SPEI, and SSMI is
geographically random and chaotic, as illustrated in Fig. 7. Furthermore, in all validation months,
ETDI, SPEI, and SSMI only provide efficacy ratios of around 50%, indicating a lack of significant
usefulness in identifying drought (Table 6). As a result, when compared to ETDI, SPEI, and SSMI,
it is clear that GTDI provides significant advantages in the field of drought monitoring. To
summarize, GTDI does not simply combine the hazard-causing index (SPEI) and the hazard-bearing
index (SSMI) as ETDI, but it can indeed capture drought occurrence in most areas, addressing the
issue of single-type drought indices' insufficient responsiveness to actual drought events.

## 383 4.3 Comparison of temporal trajectories of drought identified by

## 384 GTDI, SPEI, and SSMI

The drought identification trajectories of the integrated drought index (GTDI), single-type drought
indices (SPEI and SSMI) during the study period are depicted in Fig. 8. Out of the 850 months
spanning from March 1950 to December 2020, merely 345 months are devoid of any drought,
accounting for approximately 40.6% of the total, which contradicts our common understanding of
drought incidents. Among the 505 dry months, 409 months experience agricultural drought (SSMI,
48.1%), 356 months experience meteorological drought (SPEI, 41.9%), and 260 months (30.6%)
experience both simultaneously. GTDI identifies just 308 arid months (36.2%) out of 850 months,
which is lower than SSMI and SPEI. According to the data presented above, agricultural drought
has been the most common occurrence in the WRB over the last 70 years, followed by
meteorological drought, with GTDI identifying the fewest number of drought months.

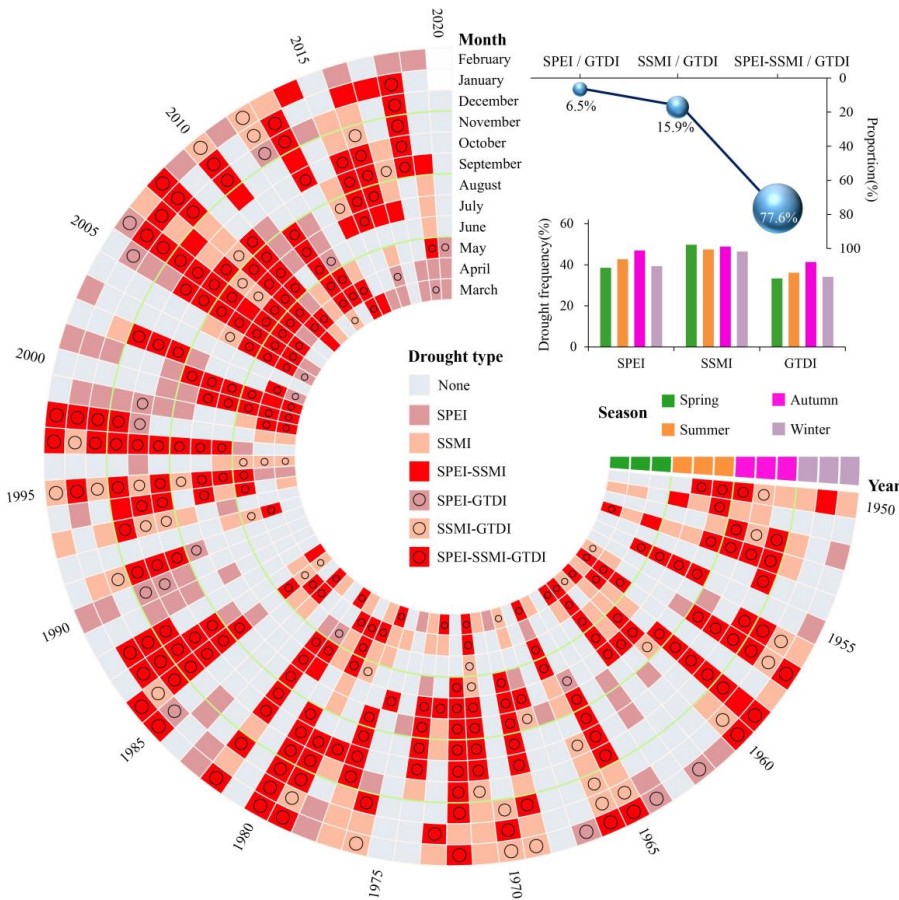


**Figure 8.** Comparison of the SPEI, SSMI and GTDI in temporal drought trajectories. "SPEI-SSMI"
means that it is recognized as a drought month by SPEI and SSMI simultaneously, and the meanings
of other drought types are similar to that.

Out of the GTDI-identified drought months, the proportion of meteorological drought

occurring alone is 6.5%, and the proportion of agricultural drought occurring alone is 15.9%,



possibly due to high temperatures, while the proportion of meteorological drought and agricultural
drought occurring simultaneously is up to 77.6%. Thus, it is clear that GTDI is closely related to the
hazard-causing index (SPEI) and the hazard-bearing index (SSMI) and is caused by both in most
cases. It corresponds to the general public's understanding of drought incidents. Furthermore,
because it is calculated by weighting SPEI and SSMI, GTDI has an advantage in depicting the
temporal gaming evolution of SPEI and SSMI. From the perspective of seasonal distribution,
meteorological drought occurs most commonly in the summer and autumn, with a frequency of
more than 40%, but less frequently in the winter and spring. At the same time, agricultural drought
(SSMI) occurs at a frequency of over 45% in all seasons, with a very similar frequency in four
seasons. The seasonal allocation mode of drought identified by GTDI is similar to that of SPEI, with
the similarity being that it occurs more frequently in summer and autumn than in winter and spring.
However, the frequency of drought determined by SPEI is slightly higher than that determined by
GTDI in each season.
The above explanation suggests that using SPEI, SSMI, and GTDI for monthly-scale drought
identification may result in various drought trajectories. Meanwhile, the GTDI is a hybrid of the
hazard-causing index (SPEI) and the hazard-bearing index (SSMI), as it has a higher overlap with
SSMI in drought trajectory, implying changes in the hazard-bearing body during the dry period,
while being closer to SPEI in drought seasonal allocation, responding to the fluctuation of hazard-
causing factors. When paired with the GTDI index reliability analysis in Section 4.2, it is concluded
that the occurrence of drought events in the Wei River Basin is still dominated by precipitation
deficiency, and the region is located in a dry location with low rainfall.





## 4.4 Comparison of spatial evolution of drought events identified by GTDI, SPEI, and SSMI

To explore the spatial development process of drought occurrences recognized by GTDI, SPEI, and

SSMI while eliminating the randomness of a single event, we selected three drought events that

lasted for a duration of 5 months for spatial evolution analysis. Fig. 9 shows the spatial evolution

processes of three drought events identified by GTDI, SPEI, and SSMI, spanning from June to

October 1982, from March to July 2000, and from September 2018 to January 2019, respectively.

Table 7 shows the drought intensity and the percentage of drought area for each month of the three

drought events.

**Table 7.** Comparison of SPEI, SSMI and GTDI in drought intensity and percentage of drought area

during three drought events

| Drought events | Year-month | Drought intensity | | | Percentage of drought area | | |
|---|---|---|---|---|---|---|---|
| | | SPEI | GTDI | SSMI | SPEI | GTDI | SSMI |
| 1982 | 1982-6 | 0.47 | 0.31 | 0.28 | 100% | 85.9% | 55.7% |
| | 1982-7 | 0.77 | 0.66 | 0.55 | 63.2% | 67.0% | 81.5% |
| | 1982-8 | 0.52 | 0.57 | 0.71 | 42.5% | 49.3% | 58.5% |
| | 1982-9 | 0.17 | 0.22 | 0.37 | 15.0% | 23.3% | 35.9% |
| | 1982-10 | 0.15 | 0.13 | 0.22 | 17.4% | 14.1% | 22.4% |
| 2000 | 2000-3 | 0.49 | 0.32 | 0.29 | 74.1% | 61.2% | 32.3% |
| | 2000-4 | 0.82 | 0.66 | 0.58 | 98.2% | 92.7% | 79.3% |
| | 2000-5 | 1.29 | 1.17 | 1.03 | 100% | 100% | 100% |
| | 2000-6 | 0.18 | 0.21 | 0.31 | 38.4% | 50.1% | 54.3% |
| | 2000-7 | 0.76 | 0.41 | 0.11 | 87.0% | 66.6% | 15.5% |
| 2018 | 2018-9 | 0.23 | 0.10 | 0.33 | 35.9% | 5.3% | 3.0% |
| | 2018-10 | 0.55 | 0.41 | 0.46 | 65.6% | 34.2% | 21.0% |
| | 2018-11 | 0.20 | 0.31 | 0.55 | 46.5% | 32.4% | 28.7% |
| | 2018-12 | 0.22 | 0.27 | 0.46 | 43.3% | 31.0% | 27.5% |
| | 2019-1 | 0.11 | 0.06 | 0.22 | 5.3% | 1.8% | 7.5% |


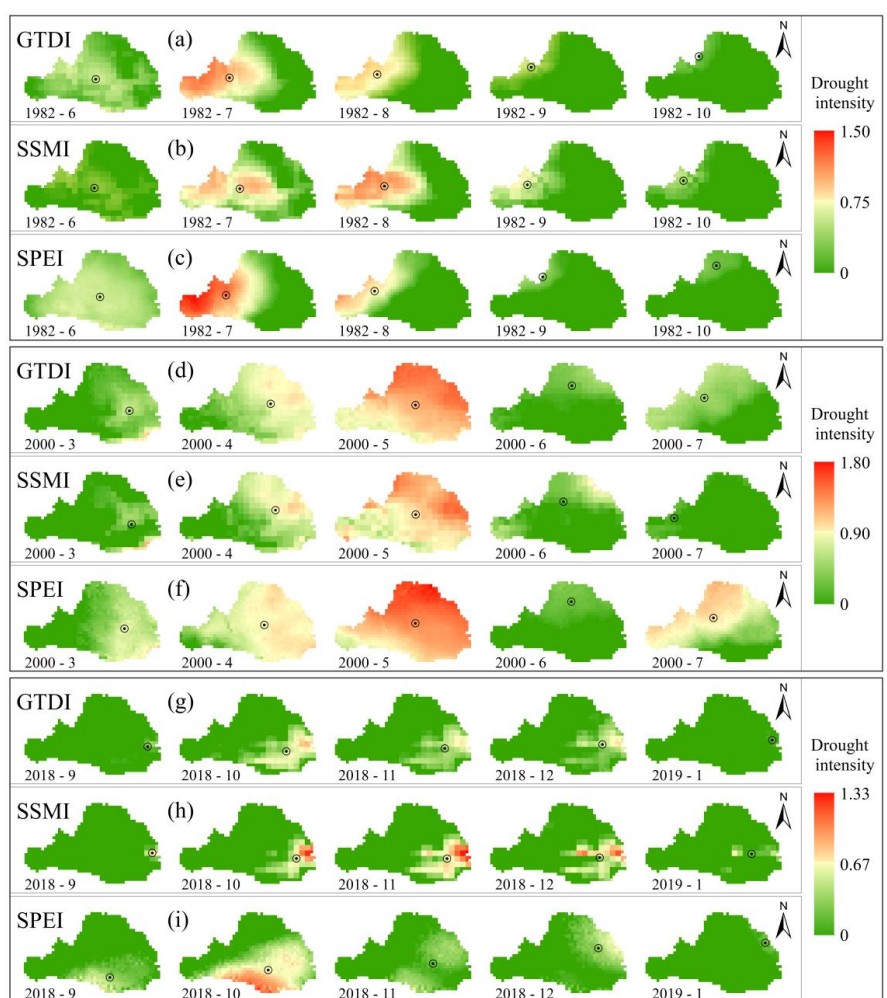

**Figure 9.** Comparison of SPEI, SSMI and GTDI in the spatial evolution of three drought events.

The black circle donates the monthly drought centroid.

Taking the 1982 drought event as an example, the meteorological drought emerges initially,

followed by a steady decrease in its impact areas (Fig. 9c). However, the overall drought intensity

increases and subsequently decreases (Table 7), and the drought centroid migrates from the WRB's

center to the northwest. It is worth noting that concurrent agricultural drought lags behind

meteorological drought. When comparing the drought geographic evolution processes identified by



SSMI and SPEI (Fig. 9b-c), the lag period is approximately one month, which is similarly observed
in the other two drought events (Fig. 9d-i). For the entire spatial evolution process of a drought event
identified by GTDI, it is clear that its spatial pattern is the result of a compromise of SPEI and SSMI,
including the migration path of the drought centroid (Fig. 9a-c), the evolution process of drought
area percentage, and drought intensity (Table 7).

From March to July 2000, the WRB experienced a fully covered drought event (Fig. 9d-f),

which began with a meteorological drought. The fusion description of SPEI and SSMI produced by
GTDI during this event, which incorporates the spatial evolution trends of SPEI and SSMI to
evaluate the current drought status at each grid point, may be observed. The value of GTDI
consistently falls between SPEI and SSMI, regardless of whether it is evaluated by the drought area
ratio, drought intensity, or drought centroid.

The 2018 drought event is the mildest of the three, but it most fully depicts the process of a

drought event from emergence to spread to eventual extinction (Fig. 9g-i). In the early stages of this
drought event, as of October 2018, the meteorological drought in the southeastern part of the WRB
was the most severe, whilst the agricultural drought was relatively negligible. In this case, the spatial
drought pattern determined by GTDI was closer to the development of hazard-causing index SPEI.
However, during the later stages of the drought event, the situation reverses and the spatial evolution
of drought begins to be dominated by the hazard-bearing index SSMI, illustrating GTDI possesses
more realistic and intelligent feature in drought identification. This also demonstrates the
importance of including game theory in this study, which has a distinct benefit in monitoring
changes in hazard-causing and bearing impacts.

Based on the foregoing, it is worth noting that the GTDI-identified spatial drought process



combines the evolutionary features of hazard-causing and bearing indices (SPEI and SSMI). In
addition, merging SPEI and SSMI via their game relationship, rather than simply putting them
together, makes GTDI a superior technique to represent the spatial and temporal evolution of
droughts. Furthermore, it has been discovered that the GTDI exhibits the gaming feature of the
drought hazard-causing and bearing index. This is evidenced by the fact that the hazard-causing
index SPEI primarily drives the early stages of drought events in the WRB, while the hazard-bearing
index SSMI primarily drives the later stages.

## 5 Conclusions

This study integrated the SPEI (meteorological index and drought hazard-causing index) and SSMI
(agricultural index and drought hazard-bearing index) to propose a game theory-based drought index
(GTDI). The integration performance and weight allocation of the GTDI were demonstrated by
evaluating the correlations with SPEI and SSMI, and comparing the integrated weight to the ETDI
(entropy theory-based drought index); the reliability of the GTDI was confirmed by the Leaf Area
Index (LAI) data; and the advancedness of the GTDI was examined by contrasting the temporal
trajectories and spatial evolution characteristics of GTDI, SPEI, and SSMI. The following are the
primary conclusions:
The single-type drought indices (SPEI and SSMI) and the integrated drought index (GTDI)
exhibit dependable spatial consistency. In all locations within the Wei River Basin during the four
seasons, there is a greatly high correlation between GTDI and SPEI. The correlation between GTDI
and SSMI is relatively weak in the winter, only reaching a high correlation in 54.8% of the basin,
while it continues to have exceptionally high correlations throughout the basin during the other three





seasons.
The entropy theory-based drought index ETDI performs worse than the GTDI, particularly
when it comes to the regional distribution of correlation coefficient homogeneity. Specially, the
game theory technique provides an integrated weight geographic distribution in the integrated index
GTDI that is consistent with the precipitation-dominated natural drought pattern. Furthermore, there
is a strong negative spatial relationship between the weight ratio of SPEI to SSMI and the average
annual precipitation in the Wei River Basin, with a correlation coefficient of -0.88. The ETDI, on
the other hand, has a very weak connection (correlation coefficient of -0.04) with the annual mean
precipitation. This indicates that the GTDI's weight distribution of SPEI and SSMI is more logical
and trustworthy.
The GTDI has superior efficacy for identifying drought when compared to the ETDI, SPEI,
and SSMI. When drought occurs, GTDI efficiently captures it with an efficacy ratio of over 70% in
all validation months, whereas ETDI, SPEI, and SSMI catch it with an efficacy ratio of
approximately 50%. In terms of drought impact, GTDI can capture drought occurrence in most
places but fails in the forest due to insufficient depth of soil surface layer measurement, whereas
ETDI, SPEI, and SSMI drought detection are geographically random and chaotic. Thus, GTDI is
expected to replace single-type drought indices to provide a more accurate portrayal of actual
drought.
The GTDI merges SPEI and SSMI via their game relationship rather than simply putting them
together, making it a superior technique to represent the spatial and temporal evolution of droughts.
Due to the GTDI is a hybrid of the hazard-causing index (SPEI) and the hazard-bearing index
(SSMI), it represents diverse drought trajectories identified by the monthly-scale SPEI and SSMI.



Specially, it has a higher overlap with SSMI in drought trajectory, implying changes in the hazard-
bearing body during the dry period, while being closer to SPEI in drought seasonal allocation,
responding to the fluctuation of hazard-causing factors. Additionally, it has been discovered that
GTDI exhibits the gaming feature of the drought hazard-causing and bearing index, having a distinct
benefit in monitoring changes in their impacts.
According to an investigation of monthly GTDI in the Wei River Basin from 1950 to 2020,
there is a growing propensity for drought, particularly since the 1990s, when the intensity and
frequency of drought in the WRB have increased significantly. Drought deterioration is most visible
in the spring, insignificant in the summer and autumn, and most areas embrace drought reduction in
the winter. Drought events in the Wei River Basin are dominated by a lack of precipitation. The
hazard-causing index SPEI dominates the early stages of a drought event, whereas the hazard-
bearing index SSMI dominates the later stages.

## Data availability

All produced data can be provided by the corresponding author upon request.

## Author contribution

Conceptualization: HZ; data curation: YZ; formal analysis: ZY; methodology: TL; investigation:
XZ; software: ZY; visualization: TL; writing – original draft: XZ; writing – review and editing: TY;
supervision: HZ; funding acquisition: HZ; project administration: HZ; validation: CX; resources:
HZ. All authors have read and agreed to publish the manuscript.

## Competing interests

The authors declare that they have no conflict of interest.





## Acknowledgments

This research is supported by the National Natural Science Foundation of China (51979005), the Natural Science Basic Research Program of Shaanxi Province (2022JC-LHJJ-03) and the Fundamental Research Funds for the Central Universities (300102293201). Our cordial thanks should be extended to the editor and anonymous reviewers for their pertinent and professional suggestions and comments which are greatly helpful for further improvement of the quality of this paper.

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
