# Peer review of "GTDI: a gaming integrated drought index implying hazard"

_Natural Hazards and Earth System Sciences, 2024_

## Community Comment (CC1)

**Review: "GTDI: a gaming integrated drought index implying hazard causing and bearing impacts changing" by Zhao et al.**

**RC1: 'Comment on nhess-2024-45', Anonymous Referee #1**

The study proposes an integrated drought index (GTDI) that combines hazard-causing (SPEI) and hazard-bearing (SSMI) indices using game theory. The GTDI is compared with an entropy-based index (ETDI) and validated against LAI data in the Wei River Basin. The key findings are that GTDI outperforms single indices and ETDI in identifying droughts spatiotemporally, and can monitor changes in hazard-causing vs bearing impacts. Overall, the manuscript is well-written and the methodology appears sound. I have a few suggestions for improvement:

We greatly appreciate the positive feedback. We would like to express our gratitude for your valuable comments and suggestions. We are grateful for the acknowledgement of the work. We would like to assure the reviewer that we will carefully address all concerns and incorporate the suggestions into an improved version of the manuscript. We are committed to enhancing the quality of our manuscript based on the reviewers' comments.

**Major suggestion:**

1. The novelty and significance of the GTDI should be highlighted more clearly in the Introduction. Discuss how it advances integrated drought indices beyond existing approaches like copula functions and entropy weighting. Articulate the unique advantages of using game theory.

**Answer 1:** We greatly appreciate the positive evaluation for this study. Indeed, we strongly agree with your opinion that emphasizing the unique advantages of using game theory will contribute to the enhancement of our manuscript. We will further highlight the advantages of game theory methods in the revised manuscript. For example:

"*Therefore, game theory is suggested for the integration of drought indices because it can comprehensively consider the opinions of each party to achieve a distribution pattern that satisfies each participant [1,2], which is superior to the entropy weight method in weight allocation, and its calculation process is simpler than copula functions.*"

[1] Lai, C., Chen, X., Chen, X., Chen, X., Wang, Z., Wu, X., and Zhao, S.: A fuzzy comprehensive evaluation model for flood risk based on the combination weight of game theory, Nat. Hazards., 77, 1243-1259, https://doi.org/10.1007/s11069-015-1645-6, 2015.

[2] Jato-Espino, D. and Ruiz-Puente, C.: Bringing Facilitated Industrial Symbiosis and Game Theory together to strengthen waste exchange in industrial parks, Sci. Total Environ., 771, 145400, https://doi.org/10.1016/j.scitotenv.2021.145400, 2021.

2. The authors compare the GTDI with the ETDI, showing that the GTDI is more accurate. However, additional drought indices like the Standardized Compound Event Indicator (SCEI) and Standardized Dry and Hot Index (SDHI) (Hao et al., 2018, 2019; Wu et al., 2020) are constructed using similar approaches. Comparing the GTDI with these indices would provide readers with valuable insights into its performance relative to state-of-the-art methods in the field.

**Answer 2:** We greatly appreciate the positive feedback for this study. We have carefully read the three references you provided and gained very useful inspiration. Thank you very much for providing us with valuable references and research ideas. Unfortunately, we found that the relevant drought indices in the references you provided mainly focus on the hot summer, which is not completely consistent with the drought period covered in this manuscript, as our research paid attention to continuous, year-round drought assessments. In any case,

your suggestions have provided us with great inspiration and thought, as well as a useful idea for our subsequent research.

3. Comment on the sensitivity of GTDI to the choice of input indices. Would the results change meaningfully if indices other than SPEI and SSMI were used? Some discussion of generalizability would be useful.

**Answer 3:** Thank you for pointing out this issue. In this study, game theory is a linear combination method that vias the data characteristics of both parties to find the optimal combination. Therefore, when the object of the game changes, such as using drought indices other than SPEI and SSMI, the weight allocation between the two parties will inevitably change accordingly, resulting in different calculation results of the integrated drought index. Your perspectives are really valuable and forward-looking, and what you point out is also ongoing work in our other related research.

4. The conclusion section could be more concise, focusing on the key findings and their implications.

**Answer 4:** Thank you for highlighting shortcomings in our manuscript. We will proceed as suggested and further condense the conclusion section to highlight the main findings and points.

**Minor suggestions:**

(Line 21-22) Include a brief description of the assessment method in the abstract.

**Answer:** Thank you very much for your comment. We apologize for the inappropriate wording. In fact, there is no assessment method here. Our "assessment" actually refers to comparing the temporal and spatial development trajectory of droughts identified by GTDI with SPEI and SSMI. The use of this word may have caused your misunderstanding. Therefore, we intend to modify this sentence as follows:

"*Furthermore, a comparative analysis is conducted on the temporal trajectories and spatial evolution of droughts identified by GTDI with SPEI and SSMI to discuss the GTDI's advancedness in monitoring changes in hazard-causing and bearing impacts.*"

(Line 24-25) Explain concisely in the abstract why the ETDI was used as a benchmark to demonstrate the GTDI's efficiency.

**Answer:** Thank you for this comment. The integrated drought index proposed in this paper is constructed based on the game theory method, while ETDI is constructed based on the entropy theory. Both methods belong to linear combination methods, and the entropy theory has been applied to the development of integrated drought indices [3]. Therefore, the comparison between ETDI and GTDI is helpful to reflect the characteristics of GTDI. To address your suggestion more comprehensively, we will briefly explain the reasons for using ETDI as a comparison for GTDI in the abstract. For example:

"*Also, the entropy theory-based drought index (ETDI) is induced to incorporate a spatial comparison to the GTDI to illustrate the rationality of gaming weight integration, as both entropy theory and game theory belong to linear combination methods in the development of the integrated drought index, and entropy theory has been applied in related research [3].*"

[3] Huang, S., Chang, J., Leng, G., and Huang, Q. Integrated index for drought assessment based on variable fuzzy set theory: a case study in the Yellow River basin, China, J. Hydrol., 527, 608-618, https://doi.org/10.1016/j.jhydrol.2015.05.032, 2015.

(Line 58-70) Consider condensing this paragraph to improve the paper's focus.

**Answer:** Thank you for your suggestion. We will improve this paragraph in the revised manuscript as suggested. For example:

*"Drought is currently categorized into four types based on distinct description objects: meteorological, agricultural, hydrological, and socioeconomic droughts [4,5]. Despite differing definitions and emphasis, meteorological drought is always regarded as the root cause of the other three types of drought [6]. In terms of the driving mechanism of drought occurrences, meteorological drought indicates the causative attribute of drought [7], whereas the other three primarily reflect the state of hazard-bearing entities. Concurrently examining the hazard-causing and hazard-bearing components of drought is essential for effective estimation and management of drought risk."*

[4] Wilhite, D.A., and Glantz, M.H.: Understanding: the drought phenomenon: the role of definitions, Water Int., 10, 111-120, 1985.

[5] Shah, D., and Mishra, V.: Integrated Drought Index (IDI) for drought monitoring and assessment in India, Water Resour. Res., 56, e2019WR026284, https://doi.org/10.1029/2019WR026284, 2020.

[6] Ma, B., Zhang, B., Jia, L., and Huang, H.: Conditional distribution selection for SPEI-daily and its revealed meteorological drought characteristics in China from 1961 to 2017, Atmos. Res., 246, 105108, https://doi.org/10.1016/j.atmosres.2020.105108, 2020.

[7] Zhang, J., Wang, J., Chen, S., Wang, M., Tang, S., and Zhao, W.: Integrated Risk Assessment of Agricultural Drought Disasters in the Major Grain-Producing Areas of Jilin Province, China, Land., 12, 160, https://doi.org/10.3390/land12010160, 2023.

(Line 76-78) Rephrase this sentence for clarity.

**Answer:** We apologize for the lack of clarity. Your suggestions are valuable for improving this manuscript. As we all know, drought events have complex causes and a wide range of impacts. A single type of drought index is often insufficient in scientifically describing the spatiotemporal evolution characteristics of droughts [8], and is insufficient to fully and objectively explain the extensiveness of the impact of drought events and the complexity of their formation [9]. Therefore, many scholars are committed to developing a comprehensive drought index from multiple dimensions. Therefore, we will rephrase this sentence as follows:

*"However, due to the complex causes and wide-ranging impacts of drought events, a single type drought index usually cannot fully and effectively reflect the spatiotemporal development process of drought events [8-9]. As a result, much effort has been expended in developing comprehensive drought indices, such as ……"*

[8] Chang, J., Li, Y., Wang, Y., and Yuan, M.: Copula-based drought risk assessment combined with an integrated index in the Wei River Basin, China, J. Hydrol., 540, 824-834, https://doi.org/10.1016/j.jhydrol.2016.06.064, 2016.

[9] Wei, H., Liu, X., Hua, W., Zhang, W., Ji, C., and Han, S.: Copula-Based Joint Drought Index Using Precipitation, NDVI, and Runoff and Its Application in the Yangtze River Basin, China, Remote Sens., 15, 4484, https://doi.org/10.3390/rs15184484, 2023.

(Line 80-81) Elaborate on why the mentioned indices struggle to distinguish between meteorological and agricultural drought influences and evaluate changes in regional patterns.

**Answer:** Thank you for this comment. The Palmer Drought Severity Index, also known as the mentioned drought index, is a drought index based on the relationship between water supply and demand [10]. Its calculation involves multiple variables such as precipitation, temperature, and soil moisture [11]. Therefore, it is difficult for us to understand the dynamic relationship between its multiple components based on this drought index alone. That is, when a drought event occurs, we cannot directly understand through PDSI whether the drought is caused by meteorological factors (precipitation shortage) or agricultural factors (soil moisture deficit). At the same time, since the dominant factors causing drought in different regions may not be consistent, it is even more difficult to obtain the spatial distribution of drought inducements using the PDSI calculated based on the relationship between water supply and demand.

[10] Palmer, W.C.: Meteorological drought, US Department of Commerce, Weather Bureau, Washington, DC, 1965.

[11] Yan, H., Wang, S. Q., Wang, J. B., Lu, H. Q., Guo, A. H., Zhu, Z. C., and Shugart, H. H. (2016). Assessing spatiotemporal variation of drought in China and its impact on agriculture during 1982–2011 by using PDSI indices and agriculture drought survey data. Journal of Geophysical Research: Atmospheres, 121(5), 2283-2298.

(Line 89-91) Copulas are an efficient tool for constructing drought indices, and the samples do not necessarily need to follow a specific probability density function. For example, empirical probability functions can effectively fit the samples. Consider modifying this paragraph accordingly.

**Answer:** Thank you for pointing out this issue, which contributes to the enhancement of our manuscript. We would like to express our sincere gratitude for pointing out the inappropriateness in the manuscript. We will choose our words carefully and modify this paragraph as follows:

*"It should be noted that copula functions are possibly reliant on the assumption that samples follow a specific probability density function [12]. However, due to the complicated interactions between the atmosphere, vegetation, soil, and groundwater, the drought does not generally meet it."*

[12] Zhang, Y., Huang, S., Huang, Q., Leng, G., Wang, H., and Wang, L.: Assessment of drought evolution characteristics based on a nonparametric and trivariate integrated drought index, J. Hydrol., 579, 124230, https://doi.org/10.1016/j.jhydrol.2019.124230, 2019.

(Line 136-137) Provide a rationale for using soil moisture data from the 0 to 10 cm surface layer, as agricultural drought indices often utilize soil moisture data from the root zone.

**Answer:** Thank you for this comment. The soil surface layer of 0 to 10 cm has a great impact on crop growth and can accurately reflect agricultural drought conditions [13]. In addition, there are many related studies using soil moisture data from 0 to 10 cm of the soil surface for agricultural drought assessment [14,15], and effective drought assessment results have been achieved. Therefore, we used soil moisture data from 0 to 10 cm to calculate the agricultural drought index SSMI in this study.

[13] Souza, A. G. S. S., Neto, A. R., and de Souza, L. L. (2021). Soil moisture-based index for agricultural drought assessment: SMADI application in Pernambuco State-Brazil. Remote Sensing of Environment, 252, 112124.

[14] Zhou, K., Li, J., Zhang, T., and Kang, A. (2021). The use of combined soil moisture data to characterize agricultural drought conditions and the relationship among different drought types in China. Agricultural Water Management, 243, 106479.

[15] Baik, J., Zohaib, M., Kim, U., Aadil, M., and Choi, M. (2019). Agricultural drought assessment based on multiple soil moisture products. Journal of arid environments, 167, 43-55.

(Line 140-141) Include details on the resampling method used.

**Answer:** Thank you for your suggestion. This study employs the bilinear interpolation method for resampling. According to your reminder, we will modify this sentence as follows:

*"Additionally, in order to facilitate calculation and analysis, precipitation, air temperature, soil moisture, and leaf area index (LAI) data were all resampled to the same spatial resolution of 0.125° using the bilinear interpolation method in this study."*

(Line 153-154) Explain the reasoning behind using SPEI-3 and SSMI-3.

**Answer:** Thank you for pointing out this issue. Drought indices at different time scales can reflect the dry and wet conditions of the study area at different time periods in the past. The 3-month drought index can reflect short- and medium-term dry and wet conditions and is more sensitive to seasonal drought, which helps us identify and analyze seasonal drought in the Wei River Basin. So, we used SPEI-3 and SSMI-3 to construct a integrated drought index GTDI with a three-month scale in this study.

(Line 206) Add a reference to support the statement.

**Answer:** Thank you for your suggestion. We will add a reference for the correlation levels of PCC as follows:

*"Table 3. The absolute value range of PCC and correlation levels [16]."*

[16] Yang, Y., and He, Y. (2022). A fault identification method based on an ensemble deep neural network and a correlation coefficient. Soft Computing, 26(18), 9199-9214.

(Line 246) Incorporate a duration threshold in the drought identification criteria.

**Answer:** Thank you for your positive feedback. It should be pointed out that the drought identification method we used, that is, a spatiotemporal continuity technique [17], is inconsistent with the research route of using drought duration to identify drought events. In the drought identification process of this study, as long as the drought index value at a grid point is lower than the drought index threshold of -1, we determine it as a drought grid point. When the total area of drought grid points in a certain month exceeds the drought area threshold, we determine that month as a drought month. Furthermore, when multiple consecutive months are determined to be drought months, if the overlapping area of drought areas in space between two adjacent consecutive drought months exceeds the drought area threshold, we determine that these two months belong to the same drought event, otherwise, they belong to different drought events. Since we are studying drought conditions at the monthly scale, the minimum drought duration of the drought event we identified is one month. Therefore, we have no way to add a drought duration threshold that does not match our drought event determination method here. But anyway, thank you very much for your valuable suggestions.

[17] Deng, C.L., She, D.X., Zhang, L.P., Zhang, Q., Liu, X., and Wang, S.X.: Characteristics of drought events using three-dimensional graph connectedness recognition method in the Yangtze River Basin, China, Trans. Chin. Soc. Agric. Eng., 37, 131-139, 2021.

We express our gratitude for your valuable input, and we assure you that all of your comments and concerns will be carefully considered and incorporated into the revised manuscript.

---

## Community Comment (CC2)

**Review:** **"GTDI: a gaming integrated drought index implying hazard causing and bearing impacts changing" by zhao et al.**

**RC2: 'Comment on nhess-2024-45', Anonymous Referee #1**

Hao, Z., Hao, F., Singh, V. P., & Zhang, X. (2018). Changes in the severity of compound drought and hot extremes over global land areas. Environmental Research Letters, 13(12), 124022. https://doi.org/10.1088/1748-9326/aaee96

Hao, Z., Hao, F., Singh, V. P., & Zhang, X. (2019). Statistical prediction of the severity of compound dry-hot events based on El Ni.o-Southern Oscillation. Journal of Hydrology, 572, 243–250. https://doi.org/10.1016/j.jhydrol.2019.03.001

Wu, X., Hao, Z., Zhang, X., Li, C., & Hao, F. (2020). Evaluation of severity changes of compound dry and hot events in China based on a multivariate multi-index approach. Journal of Hydrology, 583, 124580. https://doi.org/10.1016/j.jhydrol.2020.124580

Thank you very much for the suggestion of these references. We have already responded to these references you provided, as can be seen in the the reply (AC1) for major suggestion #2 to your first comment (RC1).

We express our gratitude for your valuable input, and we assure you that all of your comments and concerns will be carefully considered and incorporated into the revised manuscript.

---

## Author Comment (AC3)

**Review:** **"GTDI: a gaming integrated drought index implying hazard causing and bearing impacts changing"** **by Zhao et al.**

**RC3: 'Comment on nhess-2024-45', Anonymous Referee #2**

**General comments:**

This study created a new integrated drought index (GTDI) by integrating SPEI and SSMI using the game theory method. The GTDI's usefulness in drought identification was evaluated through a case study in the Wei River Basin, and the effects of hazard-causing and hazard-bearing elements on drought episodes were discovered. Overall, this drought study falls within the scope of the NHESS journal and offers useful insights for recognizing and tracking drought hazards. However, the method and analysis in the manuscript are insufficient to support several of the authors' arguments, and the general presentation of the manuscript (including some images and discusses) requires improvement. I believe that this work requires certain revisions before it can be accepted for publication. My comments can be seen below.

*We greatly appreciate the positive feedback. We would like to express our gratitude for your valuable comments and suggestions. We are grateful for the acknowledgement of the work. We would like to assure the reviewer that we will carefully address all concerns and incorporate the suggestions into an improved version of the manuscript. We are committed to enhancing the quality of our manuscript based on the reviewers' comments.*

**Specific comments:**

**Abstract**

Line 31-33, "This study surely serves as a helpful reference for the development of integrated drought indices as well as regional drought mitigation, prevention, and monitoring." I am not convinced that the integrated drought index developed in this study has practical applications for "regional drought mitigation". Please improve the phrasing in this statement.

**Answer:** *Thank you for noticing. We have improved this sentence and intend to replace it by: "This study surely serves as a helpful reference for the development of integrated drought indices as well as regional drought prevention and monitoring."*

**1. Introduction**

Line 54, "with drought occurrences becoming more frequent, intense, and extended", remove "occurrences".

**Answer:** *Thank you for this comment. We will remove "occurrences"as suggested.*

Line 89, change "has" to "had".

**Answer:** *Thank you for careful review. We will modify the text as suggested, also check and revise other similar tense issues in the text sentence by sentence.*

Line 95, according to the statements in lines 82 to 84, "comprehensive" should be replaced by "integrated" or "composite".

**Answer:** *Thank you for pointing out this issue. We will follow your suggestion to change and check for other inappropriate wording throughout the manuscript.*

Line 99-101, provide the appropriate citations for "it has been revealed that the impacts of different factors on drought, such as hazard-causing and hazard-bearing, are changing …".

**Answer:** *Thank you for this comment. According to your suggestion, we will add some citations as follows:*

*"Furthermore, it has been revealed that the impacts of different factors on drought [18,19], such as hazard-causing and hazard-bearing, are changing spatially and game-playing, necessitating the development of effective linear combination methods for measuring their spatial heterogeneity in contribution to drought."*

[18] Blauhut, V., Stahl, K., Stagge, J. H., Tallaksen, L. M., Stefano, L. D., Vogt, J. (2016). Estimating drought risk across Europe from reported drought impacts, drought indices, and vulnerability factors. Hydrol. Earth Syst. Sci., 20(7): 2779-2800.

[19] Zhang, Q., Shi, R., Singh, V. P., Xu, C., Yu, H., Fan, K., Wu, Z. (2022). Droughts across China: Drought factors, prediction and impacts. Sci. Total Environ., 803: 150018.

**2. Study area and data**

Line 124-128, the precipitation and temperature conditions in the Wei River Basin are mentioned here, however there are no corresponding subfigures in Figure 1. Please include a matching regional distribution map of precipitation and temperature in Figure 1.

**Answer:** Thank you for your suggestion. We will improve Figure 1 in the revised manuscript as suggested.

Line 134, add a citation for the DEM data.

**Answer:** Thank you for pointing out this issue. However, we have given the source of the data in accordance with the data citation requirements, which is listed in Table 1, and there is no corresponding research literature on this DEM data.

Line 135, add a citation for the precipitation and temperature dataset.

**Answer:** Thank you. We will add a citation for the precipitation and temperature dataset as follows: *"monthly precipitation and temperature dataset [20] from 1950 to 2020 with a grid size of 1 km."*

[20] Peng, S., Ding, Y., Liu, W., Li, Z. (2019). 1 km monthly temperature and precipitation dataset for China from 1901 to 2017. Earth System Science Data, 11(4), 1931-1946.

Line 136, add a citation for GLDAS_NOAH025_3H_2.0 and GLDAS_NOAH025_3H_2.1.

**Answer:** Thank you. In fact, we have given the source of the data in accordance with the data citation requirements, which can be found in Table 1 of our manuscript.

Line 138, add a citation for GLOBMAP leaf area index dataset (Version 3).

**Answer:** Thank you. We will add a citation for GLOBMAP leaf area index dataset (Version 3) as follows: *"GLOBMAP leaf area index dataset (Version 3) [21] with a period of 1981 to 2019 and a spatial resolution of 0.08°."*

[21] Liu, Y., R. Liu., J. M. Chen. (2012). Retrospective retrieval of long-term consistent global leaf area index (1981–2011) from combined AVHRR and MODIS data, J. Geophys. Res., 117, G04003, doi:10.1029/2012JG002084.

**3. Methodology**

Line 145, please put the detailed calculation procedure for SPEI in a supplementary file.

**Answer:** Thank you for your suggestion. We will put the detailed calculation procedure for SPEI in the supplementary material, which will be uploaded along with the revised manuscript.

Line 156, please put the detailed calculation procedure for SSMI in a supplementary file.

**Answer:** Thank you for your suggestion. We will put the detailed calculation procedure for SSMI in the supplementary material, which will be uploaded along with the revised manuscript.

In Section 3.2, as a comparison to the GTDI index, the calculation process of the ETDI index needs to be explained in detail.

**Answer:** Thank you for your suggestion. We totally agree with you. The calculation process of the ETDI index will be stated as suggested. However, considering the length limitation of the manuscript, we intend to put it in the supplementary material just like SPEI and SSMI.

In Table 2, the value range for "moderate drought" is wrong and should be changed to "-1.5 < Index ≤ -1.0".

**Answer:** Thank you for noticing. We will correct the value range for "moderate drought" by "-1.5 < Index ≤ -1.0".

Line 199-202, add the appropriate citations for the Pearson's correlation coefficients (PCC).

**Answer:** Thank you for your suggestion. We intend to add a citation for the Pearson's correlation coefficients (PCC) as follows:

 *"Thus, the Pearson's correlation coefficients (PCC) [22] between GTDI/ETDI with SPEI and SSMI are calculated for each grid (Eq. 6), and their correlation in different locations is explored."*

[22] Panda, P. K., Panda, R. B., Dash, P. K. (2018). The study of water quality and pearson's correlation coefficients among different physico-chemical parameters of River Salandi, Bhadrak, Odisha, India. Am. J. Water Resour., 6(4): 146-155.

Line 210, "indexes" is used incorrectly and should be replaced by "indices".

**Answer:** Thank you for noticing and we apologize for the wording error. We will correct "indexes" by "indices."

Line 230-237, the method of using Leaf Area Index (LAI) data to access the performance of the drought indices is not clearly stated. Should the comparison be between the mean values of the LAI rather than the drought indices in arid and non-arid months?

**Answer:** We apologize for the misunderstanding. As you mentioned, the comparison in the calculation is indeed the mean values of the LAI. We apologize for the lack of clarity in our description, which has caused difficulties in your reading. We intend to clarify the statement as follows:

*"If the occurrence of drought has been discovered, it can be determined by comparing the mean values of the LAI during arid months with non-arid months."*

Line 240, provide the appropriate citations for the Mann-Kendall (M-K) test.

**Answer:** Thank you for your suggestion. We intend to add a citation for the Mann-Kendall (M-K) test as follows:

*"The Mann-Kendall (M-K) test is a non-parametric statistical test method with a simple computational process [23]."*

[23] Yue, S., Wang, C. Y. (2002). Applicability of prewhitening to eliminate the influence of serial correlation on the Mann-Kendall test. Water resources research, 38(6): 4-1-4-7.

Line 247-248, it is needed to explain in more detail how to identify drought through the drought index threshold and drought area threshold, and what are the specific identification criteria?

**Answer:** Thank you for this comment. In the drought identification process of this study, as long as the drought index value at a grid point is lower than the drought index threshold of -1, we determine it as a drought grid point. When the total area of drought grid points in a certain month exceeds the drought area threshold, we determine that month as a drought month. Furthermore, when multiple consecutive months are determined to be drought months, if the overlapping area of drought areas in space between two adjacent consecutive drought months exceeds the drought area threshold, we determine that these two months belong to the same drought event, otherwise, they belong to different drought events.

**4. Results and Discussion**

The manuscript calculated four drought indices: the SPEI, SSMI, ETDI, and GTDI, but except for the GTDI, the calculations of the other three drought indices are not reflected in the results section. It is suggested that the calculation results of the SPEI, SSMI, and ETDI be placed in a supplementary file.

**Answer:** Thank you for your suggestion. In fact, the calculation results of the four drought indices (the SPEI, SSMI, ETDI, and GTDI) can be found on the Preprint nhess-2023-41 – Supplement link or on the following link: https://nhess.copernicus.org/preprints/nhess-2024-45/nhess-2024-45-supplement.zip

Line 267-269, the findings from the final month of each season were used to depict drought conditions throughout the season; why not utilize a multi-month average?

**Answer:** Thank you for this comment. Drought indices at different time scales can reflect the dry and wet conditions of the study area at different time periods in the past. This study calculated the drought index at a three-month scale, and the calculation results of each month reflect the drought conditions in the past three months. The drought index in May, August, November and February of each year just reflects the dry and wet conditions of the four seasons of spring, summer, autumn and winter in meteorology.Therefore, we use the drought index in the last month of each season to reflect the dryness and wetness of the season.

In Figure 4, "PCC" can be marked above the legend on the right, and enlarge the names of the two drought indices in each row on the left.

**Answer:** Thank you for your feedback. We will improve Figure 4 as suggested.

Line 297, "worse" is inappropriately used to describe correlation coefficients (PCC) and should be replaced by "lower."

**Answer:** Thank you for your suggestion. We will replace "worse" by "lower" here.

Line 314, "their" should be changed to "its".

**Answer:** Thank you. We will follow your suggestion to change "their" to "its" here, and we will carefully search and revise the full text for more similar questions. Thank you again for your careful review.

Line 320-321, "to contrast the weight distribution of SPEI and SSMI in ...", "allocation" may be more suitable than "distribution" here.

**Answer:** Thank you. We will modify this sentence as follows:

*"To contrast the weight distribution of SPEI and SSMI in creating the integrated drought indices GTDI and ETDI, the spatial allocation of their weight ratios (SPEI/SSMI) in the WRB is plotted, as shown in Fig. 5."*

Line 328, "comprehensive" should be replaced by "integrated".

**Answer:** Thank you for pointing out this issue. We will replace "comprehensive" by "integrated" as suggested.

Line 343, "as a consequence of comparing GTDI and ETDI, it is discovered that …", "is" should be changed to "was".

**Answer:** Thank you. According to your suggestion, we intend to change this sentence as follows: *"as a consequence of comparing GTDI and ETDI, it was discovered that …"*

Line 344-345, "which is essentially congruent with the drought generation mechanism in this basin": what is the drought generation mechanism in Wei River Basin? Please elaborate on this sentence better.

**Answer:** Thank you for this comment. In this study, drought events in the Wei River Basin are dominated by a lack of precipitation. The Standardized Precipitation Evapotranspiration Index (SPEI) is closely related to precipitation. When precipitation is low, the SPEI index will decrease, indicating an increased possibility of a meteorological drought. However, the Standardized Soil Moisture Index (SSMI) is calculated by soil moisture data, reflecting the occurrence of regional drought influenced by the change of soil moisture.

In the construction of GTDI, the weight of the meteorological drought index SPEI is slightly higher than that of the agricultural drought index SSMI, indicating that SPEI, or precipitation, dominates the changes in GTDI more, which is consistent with the occurrence of drought in the Weihe River Basin dominated by precipitation shortage. Therefore, it is mentioned that *"the game theory approach gives an integrated weight geographic distribution compatible with the precipitation-dominated natural drought pattern, which is essentially congruent with the drought generation mechanism in this basin."*

Figures 6 and 7 can be combined into one figure.

**Answer:** Thank you for your feedback. We agree with your suggestion to a certain extent, but considering that it is not easy to arrange the sub-figures after combining the two figures, we still hope to draw both Figure 6 and Figure 7 separately, because it is also more convenient to compare the comprehensive drought index GTDI focused in this manuscript with the other three drought indices.

Line 371, "the soil moisture data used in this study is only 0 to 10cm of soil surface layer", "is" should be changed to "are".

**Answer:** Thank you for your suggestion. We will change "is" to "are" as suggested.

Figure 9 needs to be streamlined, as the year-month labeling is somewhat repetitive. It is suggested that the three drought indices be marked with only one year-month label under each drought event image.

**Answer:** Thank you for this comment. We will follow your suggestion to modify Figure 9.

Line 463-466, "In addition" and "Furthermore" are repeated, "in addition" can be removed.

**Answer:** Thank you for your feedback. We will remove "in addition" as suggested.

**5. Conclusions**

Line 482, add " between "correlation" and "in".

**Answer:** Thank you for this comment. According to the place you pointed out, we intend to modify it as follows:

*"The correlation between GTDI and SSMI is relatively weak in the winter, only reaching a high correlation level in 54.8% of the basin,"*

Line 492, the same as the comment for line 320-321, "allocation" may be more suitable than "distribution" here.

**Answer:** Thank you. We will  modify this sentence as follows:

*"This indicates that the GTDI's weight allocation of SPEI and SSMI is more logical and trustworthy."*

Line 511-513, the evolution trend of the GTDI is first presented in the results section, why don't authors summarize the findings of this part in the first conclusion?

**Answer:** Thank you for pointing out this issue. This part of the conclusion is not the key finding of this study. It is just a summary of the evolution trend of drought in the Wei River Basin, based on the evolution trend of GTDI in recent decades and the main findings of this study. Therefore, we put it at the end of the conclusion.

We sincerely appreciate your positive feedback and the valuable insights you have provided. Your comments and concerns have been duly noted, and we are committed to addressing each of them in the revised version of the manuscript.

---

## Author Comment (AC6)

**Review:** **"GTDI: a gaming integrated drought index implying hazard causing and bearing impacts changing"** by Zhao et al.

**RC6: 'Comment on nhess-2024-45', Anonymous Referee #1**

We would like to express our gratitude for your valuable comments and suggestions. In response to your Major suggestion #4, we have further supplemented its corresponding modification example to improve our response to your suggestion.

**Major suggestion:**

**4.** The conclusion section could be more concise, focusing on the key findings and their implications.

**Answer 4:** Thank you for highlighting shortcomings in our manuscript. We will proceed as suggested and further condense the conclusion section to highlight the main findings and points. For example:

*"This study integrated the SPEI (meteorological index and drought hazard-causing index) and SSMI (agricultural index and drought hazard-bearing index) to propose a game theory-based drought index (GTDI). The integration performance and weight allocation of the GTDI were demonstrated by evaluating the correlations with SPEI and SSMI, and comparing the integrated weight to the ETDI (entropy theory-based drought index); the reliability of the GTDI was confirmed by the Leaf Area Index (LAI) data; and the advancedness of the GTDI was examined by contrasting the temporal trajectories and spatial evolution characteristics of GTDI, SPEI, and SSMI. The following are the primary conclusions:*

*The single-type drought indices (SPEI and SSMI) and the integrated drought index (GTDI) exhibit dependable spatial consistency. The entropy theory-based drought index ETDI performs worse than the GTDI, particularly when it comes to the regional distribution of correlation coefficient homogeneity. Specially, the game theory technique provides an integrated weight geographic distribution in the integrated index GTDI that is consistent with the precipitation-dominated natural drought pattern, as there is a strong negative spatial relationship between the weight ratio of SPEI to SSMI and the average annual precipitation in the Wei River Basin. The ETDI, on the other hand, has a very weak connection with the annual mean precipitation. This indicates that the GTDI's weight distribution of SPEI and SSMI is more logical and trustworthy.*

*The GTDI has superior efficacy for identifying drought when compared to the ETDI, SPEI, and SSMI, as the GTDI efficiently captures drought with an efficacy ratio of over 70% in all validation months, whereas the ETDI, SPEI, and SSMI catch it with an efficacy ratio of approximately 50%. Thus, GTDI is expected to replace single-type drought indices to provide a more accurate portrayal of actual drought.*

*The GTDI merges SPEI and SSMI via their game relationship rather than simply putting them together, making it a superior technique to represent the spatial and temporal evolution of droughts. Specially, it has a higher overlap with SSMI in drought trajectory, implying changes in the hazard-bearing body during the dry period, while being closer to SPEI in drought seasonal allocation, responding to the fluctuation of hazard-causing factors.*

*Additionally, it has been discovered that GTDI exhibits the gaming feature of the drought hazard-causing and bearing index, having a distinct benefit in monitoring changes in their impacts. The hazard-causing index SPEI dominates the early stages of a drought event, whereas the hazard-bearing index SSMI dominates the later stages."*

We express our gratitude for your valuable input, and we assure you that all of your comments and concerns will be carefully considered and incorporated into the revised manuscript.

---

## Author Response (AR2)

**Author's Response to peer-reviews for nhess-2024-45**

"GTDI: a gaming integrated drought index implying hazard causing and bearing impacts changing" by Zhao et al.

First, we want to express our gratitude to anonymous reviewers for taking their time to review our manuscript. Their feedback is very encouraging and helpful in the enhancement of the manuscript. We have carefully studied, considered and responded to all comments point-by-point as follows. For clarity, all comments are given in black and responses are given in blue text.

**Review 1**
**Comment on nhess-2024-45, Anonymous Referee #1**

The study proposes an integrated drought index (GTDI) that combines hazard-causing (SPEI) and hazard-bearing (SSMI) indices using game theory. The GTDI is compared with an entropy-based index (ETDI) and validated against LAI data in the Wei River Basin. The key findings are that GTDI outperforms single indices and ETDI in identifying droughts spatiotemporally, and can monitor changes in hazard-causing vs bearing impacts. Overall, the manuscript is well-written and the methodology appears sound. I have a few suggestions for improvement:

We greatly appreciate the positive feedback. We would like to express our gratitude for your valuable comments and suggestions. We are grateful for the acknowledgement of the work. We are committed to enhancing the quality of our manuscript based on the reviewers' comments.

**Major suggestion:**

1. The novelty and significance of the GTDI should be highlighted more clearly in the Introduction. Discuss how it advances integrated drought indices beyond existing approaches like copula functions and entropy weighting. Articulate the unique advantages of using game theory.

**Answer 1:** We greatly appreciate the positive evaluation for this study. Indeed, we strongly agree with your opinion that emphasizing the unique advantages of using game theory will contribute to the enhancement of our manuscript. We haved further highlighted the advantages of game theory methods in the revised manuscript as follows:

Lines 101–105: *"Therefore, game theory is suggested for the integration of drought indices because it can comprehensively consider the opinions of each party to achieve a distribution pattern that satisfies each participant [1,2], which is superior to the entropy weight method in weight allocation, and its calculation process is simpler than copula functions."*

[1] Lai, C., Chen, X., Chen, X., Chen, X., Wang, Z., Wu, X., and Zhao, S.: A fuzzy comprehensive evaluation model for flood risk based on the combination weight of game theory, Nat. Hazards., 77, 1243-1259, https://doi.org/10.1007/s11069-015-1645-6, 2015.

[2] Jato-Espino, D. and Ruiz-Puente, C.: Bringing Facilitated Industrial Symbiosis and Game Theory together to strengthen waste exchange in industrial parks, Sci. Total Environ., 771, 145400, https://doi.org/10.1016/j.scitotenv.2021.145400, 2021.

2. The authors compare the GTDI with the ETDI, showing that the GTDI is more accurate. However, additional drought indices like the Standardized Compound Event Indicator (SCEI) and Standardized Dry and Hot Index (SDHI) (Hao et al., 2018, 2019; Wu et al., 2020)

are constructed using similar approaches. Comparing the GTDI with these indices would provide readers with valuable insights into its performance relative to state-of-the-art methods in the field.

Hao, Z., Hao, F., Singh, V. P., & Zhang, X. (2018). Changes in the severity of compound drought and hot extremes over global land areas. Environmental Research Letters, 13(12), 124022. https://doi.org/10.1088/1748-9326/aaee96

Hao, Z., Hao, F., Singh, V. P., & Zhang, X. (2019). Statistical prediction of the severity of compound dry-hot events based on El Ni.o-Southern Oscillation. Journal of Hydrology, 572, 243–250. https://doi.org/10.1016/j.jhydrol.2019.03.001

Wu, X., Hao, Z., Zhang, X., Li, C., & Hao, F. (2020). Evaluation of severity changes of compound dry and hot events in China based on a multivariate multi-index approach. Journal of Hydrology, 583, 124580. https://doi.org/10.1016/j.jhydrol.2020.124580

**Answer 2:** We greatly appreciate the positive feedback for this study. We have carefully read the three references you provided and gained very useful inspiration. Thank you very much for providing us with valuable references and research ideas. Unfortunately, we found that the relevant drought indices in the references you provided mainly focus on the hot summer, which is not completely consistent with the drought period covered in this manuscript, as our research paid attention to continuous, year-round drought assessments. In any case, your suggestions have provided us with great inspiration and thought, as well as a useful idea for our subsequent research.

3. Comment on the sensitivity of GTDI to the choice of input indices. Would the results change meaningfully if indices other than SPEI and SSMI were used? Some discussion of generalizability would be useful.

**Answer 3:** Thank you for pointing out this issue. In this study, game theory is a linear combination method that vias the data characteristics of both parties to find the optimal combination. Therefore, when the object of the game changes, such as using drought indices other than SPEI and SSMI, the weight allocation between the two parties will inevitably change accordingly, resulting in different calculation results of the integrated drought index. Your perspectives are really valuable and forward-looking, and what you point out is also ongoing work in our other related research.

4. The conclusion section could be more concise, focusing on the key findings and their implications.

**Answer 4:** Thank you for highlighting shortcomings in our manuscript. We have condensed the conclusion section to highlight the main findings and points as follows:

Lines 480–510: "*This study integrated the SPEI (meteorological index and drought hazard-causing index) and SSMI (agricultural index and drought hazard-bearing index) to propose a game theory-based drought index (GTDI). The integration performance and weight allocation of the GTDI were demonstrated by evaluating the correlations with SPEI and SSMI, and comparing the integrated weight to the ETDI (entropy theory-based drought index); the reliability of the GTDI was confirmed by the Leaf Area Index (LAI) data; and the advancedness of the GTDI was examined by contrasting the temporal trajectories and spatial evolution characteristics of GTDI, SPEI, and SSMI. The following are the primary conclusions:*

*The single-type drought indices (SPEI and SSMI) and the integrated drought index (GTDI) exhibit dependable spatial consistency. The entropy theory-based drought index ETDI performs worse than the GTDI, particularly when it comes to the regional distribution of correlation coefficient homogeneity. Specially, the game theory technique provides an integrated weight geographic distribution in the integrated index GTDI that is consistent with*

*the precipitation-dominated natural drought pattern, as there is a strong negative spatial relationship between the weight ratio of SPEI to SSMI and the average annual precipitation in the Wei River Basin. The ETDI, on the other hand, has a very weak connection with the annual mean precipitation. This indicates that the GTDI's weight allocation of SPEI and SSMI is more logical and trustworthy.*

*The GTDI has superior efficacy for identifying drought when compared to the ETDI, SPEI, and SSMI, as the GTDI efficiently captures drought with an efficacy ratio of over 70% in all validation months, whereas the ETDI, SPEI, and SSMI catch it with an efficacy ratio of approximately 50%. Thus, GTDI is expected to replace single-type drought indices to provide a more accurate portrayal of actual drought.*

*The GTDI merges SPEI and SSMI via their game relationship rather than simply putting them together, making it a superior technique to represent the spatial and temporal evolution of droughts. Specially, it has a higher overlap with SSMI in drought trajectory, implying changes in the hazard-bearing body during the dry period, while being closer to SPEI in drought seasonal allocation, responding to the fluctuation of hazard-causing factors.*

*Additionally, it has been discovered that GTDI exhibits the gaming feature of the drought hazard-causing and bearing index, having a distinct benefit in monitoring changes in their impacts. The hazard-causing index SPEI dominates the early stages of a drought event, whereas the hazard-bearing index SSMI dominates the later stages."*

**Minor suggestions:**

(Line 21-22) Include a brief description of the assessment method in the abstract.

**Answer:** Thank you very much for your comment. We apologize for the inappropriate wording. In fact, there is no assessment method here. Our "assessment" actually refers to comparing the temporal and spatial development trajectory of droughts identified by GTDI with SPEI and SSMI. The use of this word may have caused your misunderstanding. Therefore, we have modified this sentence as follows:

Lines 19–22: *"Furthermore, a comparative analysis is conducted on the temporal trajectories and spatial evolution of droughts identified by GTDI with SPEI and SSMI to discuss the GTDI's advancedness in monitoring changes in hazard-causing and bearing impacts."*

(Line 24-25) Explain concisely in the abstract why the ETDI was used as a benchmark to demonstrate the GTDI's efficiency.

**Answer:** Thank you for this comment. The integrated drought index proposed in this paper is constructed based on the game theory method, while ETDI is constructed based on the entropy theory. Both methods belong to linear combination methods, and the entropy theory has been applied to the development of integrated drought indices [3]. Therefore, the comparison between ETDI and GTDI is helpful to reflect the characteristics of GTDI. To address your suggestion more comprehensively, we have briefly explained the reasons for using ETDI as a comparison for GTDI in the abstract as follows:

Lines 22–25: *"Also, the entropy theory-based drought index (ETDI) is induced to incorporate a spatial comparison to the GTDI to illustrate the rationality of gaming weight integration, as both entropy theory and game theory belong to linear combination methods in the development of the integrated drought index, and entropy theory has been applied in related research."*

(Line 58-70) Consider condensing this paragraph to improve the paper's focus.

**Answer:** Thank you for your suggestion. We have improved this paragraph in the revised manuscript as suggested:

Lines 61–68: *"Drought is currently categorized into four types based on distinct description objects: meteorological, agricultural, hydrological, and socioeconomic droughts [4,5]. Despite differing definitions and emphasis, meteorological drought is always regarded as the root cause of the other three types of drought [6]. In terms of the driving mechanism of drought occurrences, meteorological drought indicates the causative attribute of drought [7], whereas the other three primarily reflect the state of hazard-bearing entities. Concurrently examining the hazard-causing and hazard-bearing components of drought is essential for effective estimation and management of drought risk."*

[4] Wilhite, D.A., and Glantz, M.H.: Understanding: the drought phenomenon: the role of definitions, Water Int., 10, 111-120, 1985.

[5] Shah, D., and Mishra, V.: Integrated Drought Index (IDI) for drought monitoring and assessment in India, Water Resour. Res., 56, e2019WR026284, https://doi.org/10.1029/2019WR026284, 2020.

[6] Ma, B., Zhang, B., Jia, L., and Huang, H.: Conditional distribution selection for SPEI-daily and its revealed meteorological drought characteristics in China from 1961 to 2017, Atmos. Res., 246, 105108, https://doi.org/10.1016/j.atmosres.2020.105108, 2020.

[7] Zhang, J., Wang, J., Chen, S., Wang, M., Tang, S., and Zhao, W.: Integrated Risk Assessment of Agricultural Drought Disasters in the Major Grain-Producing Areas of Jilin Province, China, Land., 12, 160, https://doi.org/10.3390/land12010160, 2023.

(Line 76-78) Rephrase this sentence for clarity.

**Answer:** We apologize for the lack of clarity. Your suggestions are valuable for improving this manuscript. As we all know, drought events have complex causes and a wide range of impacts. A single type of drought index is often insufficient in scientifically describing the spatiotemporal evolution characteristics of droughts [8], and is insufficient to fully and objectively explain the extensiveness of the impact of drought events and the complexity of their formation [9]. Therefore, many scholars are committed to developing a comprehensive drought index from multiple dimensions. Therefore, we have rephrased this sentence as follows:

Lines 74–77: *"However, due to the complex causes and wide-ranging impacts of drought events, a single type drought index usually cannot fully and effectively reflect the spatiotemporal development process of drought events [8-9]. As a result, much effort has been expended in developing comprehensive drought indices, such as ……"*

[8] Chang, J., Li, Y., Wang, Y., and Yuan, M.: Copula-based drought risk assessment combined with an integrated index in the Wei River Basin, China, J. Hydrol., 540, 824-834, https://doi.org/10.1016/j.jhydrol.2016.06.064, 2016.

[9] Wei, H., Liu, X., Hua, W., Zhang, W., Ji, C., and Han, S.: Copula-Based Joint Drought Index Using Precipitation, NDVI, and Runoff and Its Application in the Yangtze River Basin, China, Remote Sens., 15, 4484, https://doi.org/10.3390/rs15184484, 2023.

(Line 80-81) Elaborate on why the mentioned indices struggle to distinguish between meteorological and agricultural drought influences and evaluate changes in regional patterns.

**Answer:** Thank you for this comment. The Palmer Drought Severity Index, also known as the mentioned drought index, is a drought index based on the relationship between water supply and demand [10]. Its calculation involves multiple variables such as precipitation, temperature, and soil moisture [11]. Therefore, it is difficult for us to understand the dynamic relationship between its multiple components based on this drought index alone. That is, when a drought event occurs, we cannot directly understand through PDSI whether

the drought is caused by meteorological factors (precipitation shortage) or agricultural factors (soil moisture deficit). At the same time, since the dominant factors causing drought in different regions may not be consistent, it is even more difficult to obtain the spatial distribution of drought inducements using the PDSI calculated based on the relationship between water supply and demand.

[10] Palmer, W.C.: Meteorological drought, US Department of Commerce, Weather Bureau, Washington, DC, 1965.

[11] Yan, H., Wang, S. Q., Wang, J. B., Lu, H. Q., Guo, A. H., Zhu, Z. C., and Shugart, H. H. (2016). Assessing spatiotemporal variation of drought in China and its impact on agriculture during 1982–2011 by using PDSI indices and agriculture drought survey data. Journal of Geophysical Research: Atmospheres, 121(5), 2283-2298.

(Line 89-91) Copulas are an efficient tool for constructing drought indices, and the samples do not necessarily need to follow a specific probability density function. For example, empirical probability functions can effectively fit the samples. Consider modifying this paragraph accordingly.

**Answer:** Thank you for pointing out this issue, which contributes to the enhancement of our manuscript. We would like to express our sincere gratitude for pointing out the inappropriateness in the manuscript. We have chosen our words carefully and modified this paragraph as follows:

Lines 87–90: *"It should be noted that copula functions are possibly reliant on the assumption that samples follow a specific probability density function [12]. However, due to the complicated interactions between the atmosphere, vegetation, soil, and groundwater, the drought does not generally meet it."*

[12] Zhang, Y., Huang, S., Huang, Q., Leng, G., Wang, H., and Wang, L.: Assessment of drought evolution characteristics based on a nonparametric and trivariate integrated drought index, J. Hydrol., 579, 124230, https://doi.org/10.1016/j.jhydrol.2019.124230, 2019.

(Line 136-137) Provide a rationale for using soil moisture data from the 0 to 10 cm surface layer, as agricultural drought indices often utilize soil moisture data from the root zone.

**Answer:** Thank you for this comment. The soil surface layer of 0 to 10 cm has a great impact on crop growth and can accurately reflect agricultural drought conditions [13]. In addition, there are many related studies using soil moisture data from 0 to 10 cm of the soil surface for agricultural drought assessment [14,15], and effective drought assessment results have been achieved. Therefore, we used soil moisture data from 0 to 10 cm to calculate the agricultural drought index SSMI in this study.

[13] Souza, A. G. S. S., Neto, A. R., and de Souza, L. L. (2021). Soil moisture-based index for agricultural drought assessment: SMADI application in Pernambuco State-Brazil. Remote Sensing of Environment, 252, 112124.

[14] Zhou, K., Li, J., Zhang, T., and Kang, A. (2021). The use of combined soil moisture data to characterize agricultural drought conditions and the relationship among different drought types in China. Agricultural Water Management, 243, 106479.

[15] Baik, J., Zohaib, M., Kim, U., Aadil, M., and Choi, M. (2019). Agricultural drought assessment based on multiple soil moisture products. Journal of arid environments, 167, 43-55.

(Line 140-141) Include details on the resampling method used.

**Answer:** Thank you for your suggestion. This study employs the bilinear interpolation method for resampling. According to your reminder, we have modified this sentence as follows:

Lines 139–142: *"Additionally, in order to facilitate calculation and analysis, precipitation, air temperature, soil moisture, and leaf area index (LAI) data were all resampled to the same spatial resolution of 0.125° using the bilinear interpolation method in this study."*

(Line 153-154) Explain the reasoning behind using SPEI-3 and SSMI-3.

**Answer:** Thank you for pointing out this issue. Drought indices at different time scales can reflect the dry and wet conditions of the study area at different time periods in the past. The 3-month drought index can reflect short- and medium-term dry and wet conditions and is more sensitive to seasonal drought, which helps us identify and analyze seasonal drought in the Wei River Basin. So, we used SPEI-3 and SSMI-3 to construct a integrated drought index GTDI with a three-month scale in this study.

(Line 206) Add a reference to support the statement.

**Answer:** Thank you for your suggestion. We have added a reference for the correlation levels of PCC as follows:

Line 209: *"Table 3. The absolute value range of PCC and correlation levels [16]."*

[16] Yang, Y., and He, Y. (2022). A fault identification method based on an ensemble deep neural network and a correlation coefficient. Soft Computing, 26(18), 9199-9214.

(Line 246) Incorporate a duration threshold in the drought identification criteria.

**Answer:** Thank you for your positive feedback. It should be pointed out that the drought identification method we used, that is, a spatiotemporal continuity technique [17], is inconsistent with the research route of using drought duration to identify drought events. In the drought identification process of this study, as long as the drought index value at a grid point is lower than the drought index threshold of -1, we determine it as a drought grid point. When the total area of drought grid points in a certain month exceeds the drought area threshold, we determine that month as a drought month. Furthermore, when multiple consecutive months are determined to be drought months, if the overlapping area of drought areas in space between two adjacent consecutive drought months exceeds the drought area threshold, we determine that these two months belong to the same drought event, otherwise, they belong to different drought events. Since we are studying drought conditions at the monthly scale, the minimum drought duration of the drought event we identified is one month. Therefore, we have no way to add a drought duration threshold that does not match our drought event determination method here. But anyway, thank you very much for your valuable suggestions.

[17] Deng, C.L., She, D.X., Zhang, L.P., Zhang, Q., Liu, X., and Wang, S.X.: Characteristics of drought events using three-dimensional graph connectedness recognition method in the Yangtze River Basin, China, Trans. Chin. Soc. Agric. Eng., 37, 131-139, 2021.

**Review 2**

**Comment on nhess-2024-45, Anonymous Referee #2**

**General comments:**

This study created a new integrated drought index (GTDI) by integrating SPEI and SSMI using the game theory method. The GTDI's usefulness in drought identification was evaluated through a case study in the Wei River Basin, and the effects of hazard-causing and hazard-bearing elements on drought episodes were discovered. Overall, this drought study falls within the scope of the NHESS journal and offers useful insights for recognizing and tracking drought hazards. However, the method and analysis in the manuscript are insufficient to support several of the authors' arguments, and the general presentation of the manuscript (including some images and discusses) requires improvement. I believe that this work requires certain revisions before it can be accepted for publication. My comments can be seen below.

We greatly appreciate the positive feedback. We would like to express our gratitude for your valuable comments and suggestions. We are grateful for the acknowledgement of the work. We are committed to enhancing the quality of our manuscript based on the reviewers' comments.

**Specific comments:**

**Abstract**

Line 31-33, "This study surely serves as a helpful reference for the development of integrated drought indices as well as regional drought mitigation, prevention, and monitoring." I am not convinced that the integrated drought index developed in this study has practical applications for "regional drought mitigation". Please improve the phrasing in this statement.

**Answer:** Thank you for noticing. We have improved this sentence by:

Lines 33–35: *"This study surely serves as a helpful reference for the development of integrated drought indices as well as regional drought prevention and monitoring."*

**1. Introduction**

Line 54, "with drought occurrences becoming more frequent, intense, and extended", remove "occurrences".

**Answer:** Thank you for this comment. We have removed "occurrences"as suggested. (Line 57)

Line 89, change "has" to "had".

**Answer:** Thank you for careful review. We have modified the text as suggested (Line 87), also checked and revised other similar tense issues in the text sentence by sentence.

Line 95, according to the statements in lines 82 to 84, "comprehensive" should be replaced by "integrated" or "composite".

**Answer:** Thank you for pointing out this issue. We have replaced "A comprehensive" with " An integrated" as suggested (Line 93). And we have followed your suggestion to change and check for other inappropriate wording throughout the manuscript.

Line 99-101, provide the appropriate citations for "it has been revealed that the impacts of different factors on drought, such as hazard-causing and hazard-bearing, are changing …".

**Answer:** Thank you for this comment. According to your suggestion, we have added some citations as follows:

Lines 97–101: "*Furthermore, it has been revealed that the impacts of different factors on drought [18,19], such as hazard-causing and hazard-bearing, are changing spatially and game-playing, necessitating the development of effective linear combination methods for measuring their spatial heterogeneity in contribution to drought.*"

[18] Blauhut, V., Stahl, K., Stagge, J. H., Tallaksen, L. M., Stefano, L. D., Vogt, J. (2016). Estimating drought risk across Europe from reported drought impacts, drought indices, and vulnerability factors. Hydrol. Earth Syst. Sci., 20(7): 2779-2800.

[19] Zhang, Q., Shi, R., Singh, V. P., Xu, C., Yu, H., Fan, K., Wu, Z. (2022). Droughts across China: Drought factors, prediction and impacts. Sci. Total Environ., 803: 150018.

**2. Study area and data**

Line 124-128, the precipitation and temperature conditions in the Wei River Basin are mentioned here, however there are no corresponding subfigures in Figure 1. Please include a matching regional distribution map of precipitation and temperature in Figure 1.

**Answer:** Thank you for your suggestion. We have improved Figure 1 in the revised manuscript as suggested.

[Figure]

Figure 1. A map of the Wei River Basin.

Line 134, add a citation for the DEM data.

**Answer:** Thank you for pointing out this issue. However, we have given the source of the data in accordance with the data citation requirements, which is listed in Table 1, and there is no corresponding research literature on this DEM data.

Line 135, add a citation for the precipitation and temperature dataset.

**Answer:** Thank you. We have added a citation for the precipitation and temperature dataset as follows:

Lines 134–135: "*monthly precipitation and temperature dataset [20] from 1950 to 2020 with a grid size of 1 km.*"

[20] Peng, S., Ding, Y., Liu, W., Li, Z. (2019). 1 km monthly temperature and precipitation dataset for China from 1901 to 2017. Earth System Science Data, 11(4), 1931-1946.

Line 136, add a citation for GLDAS_NOAH025_3H_2.0 and GLDAS_NOAH025_3H_2.1.

**Answer:** Thank you. In fact, we have given the source of the data in accordance with the data citation requirements, which can be found in Table 1 of our manuscript.

Line 138, add a citation for GLOBMAP leaf area index dataset (Version 3).

**Answer:** Thank you. We have added a citation for GLOBMAP leaf area index dataset (Version 3) as follows:

Lines 138–139: "*GLOBMAP leaf area index dataset (Version 3) with a period of 1981 to 2019 and a spatial resolution of 0.08°[21].*"

[21] Liu, Y., R. Liu., J. M. Chen. (2012). Retrospective retrieval of long-term consistent global leaf area index (1981–2011) from combined AVHRR and MODIS data, J. Geophys. Res., 117, G04003, doi:10.1029/2012JG002084.

**3. Methodology**

Line 145, please put the detailed calculation procedure for SPEI in a supplementary file.

**Answer:** Thank you for your suggestion. We have put the detailed calculation procedure for SPEI in Supplement S1, which has been uploaded along with the revised manuscript.

Line 156, please put the detailed calculation procedure for SSMI in a supplementary file.

**Answer:** Thank you for your suggestion. We have put the detailed calculation procedure for SSMI in Supplement S2, which has been uploaded along with the revised manuscript.

In Section 3.2, as a comparison to the GTDI index, the calculation process of the ETDI index needs to be explained in detail.

**Answer:** Thank you for your suggestion. We totally agree with you. The calculation process of the ETDI index has been stated as suggested. However, considering the length limitation of the manuscript, we have put it in Supplement S3.

In Table 2, the value range for "moderate drought" is wrong and should be changed to "-1.5 < Index ≤ -1.0".

**Answer:** Thank you for noticing. We have corrected the value range for "moderate drought" by "-1.5 < Index ≤ -1.0".

Line 199-202, add the appropriate citations for the Pearson's correlation coefficients (PCC).

**Answer:** Thank you for your suggestion. We have added a citation for the Pearson's correlation coefficients (PCC) as follows:

Lines 203–205: *"Thus, the Pearson's correlation coefficients (PCC) [22] between GTDI/ETDI with SPEI and SSMI are calculated for each grid (Eq. 6), and their correlation in different locations is explored."*

[22] Panda, P. K., Panda, R. B., Dash, P. K. (2018). The study of water quality and pearson's correlation coefficients among different physico-chemical parameters of River Salandi, Bhadrak, Odisha, India. Am. J. Water Resour., 6(4): 146-155.

Line 210, "indexes" is used incorrectly and should be replaced by "indices".

**Answer:** Thank you for noticing and we apologize for the wording error. We have corrected "indexes" by "indices." (Line 213)

Line 230-237, the method of using Leaf Area Index (LAI) data to access the performance of the drought indices is not clearly stated. Should the comparison be between the mean values of the LAI rather than the drought indices in arid and non-arid months?

**Answer:** We apologize for the misunderstanding. As you mentioned, the comparison in the calculation is indeed the mean values of the LAI. We apologize for the lack of clarity in our description, which has caused difficulties in your reading. We have clarified the statement as follows:

Lines 233–239: *"If the occurrence of drought has been discovered, it can be determined by comparing the mean values of the LAI during arid months with non-arid months. The specific process is as follows:*

$$\begin{cases} M_{d,i} = \dfrac{\sum_{j=1}^{m} I_{i,j}}{m} \\[2em] M_{n,i} = \dfrac{\sum_{l=1}^{n} I_{i,l}}{n} \end{cases} \tag{7}$$

$$R_i = \begin{cases} 1, M_{d,i} < M_{n,i} \\ 0, M_{d,i} \geq M_{n,i} \end{cases} \tag{8}$$

*where $M_{d,i}$ and $M_{n,i}$ represent the average values of the LAI in the i-th grid during arid and non-arid months, respectively; m and n are the number of arid and non-arid months, respectively; $I_{i,j}$ and $I_{i,l}$ represent the value of the LAI of the i-th grid during the j-th arid month and the l-th non-arid month, respectively; $R_i$ represents the drought recognition performance of the drought index in the i-th grid, with a value of 1 indicating fine and 0 indicating poor."*

Line 240, provide the appropriate citations for the Mann-Kendall (M-K) test.

**Answer:** Thank you for your suggestion. We have added a citation for the Mann-Kendall (M-K) test as follows:

Lines 242–243: *"The Mann-Kendall (M-K) test is a non-parametric statistical test method with a simple computational process [23]."*

[23] Yue, S., Wang, C. Y. (2002). Applicability of prewhitening to eliminate the influence of serial correlation on the Mann-Kendall test. Water resources research, 38(6): 4-1-4-7.

Line 247-248, it is needed to explain in more detail how to identify drought through the drought index threshold and drought area threshold, and what are the specific identification criteria?

**Answer:** Thank you for this comment. We have added a paragraph to explain how to identify drought through the drought index threshold and drought area threshold in the revised manuscript as follows:

Lines 252–258: *"Briefly, as long as the drought index value at a grid point is lower than the drought index threshold of -1, we determine it as a drought grid point. When the total area of drought grid points in a certain month exceeds the drought area threshold, we determine that month as a drought month. Furthermore, when multiple consecutive months are determined to be drought months, if the overlapping area of drought areas in space between two adjacent consecutive drought months exceeds the drought area threshold, we determine that these two months belong to the same drought event, otherwise, they belong to different drought events."*

**4. Results and Discussion**

The manuscript calculated four drought indices: the SPEI, SSMI, ETDI, and GTDI, but except for the GTDI, the calculations of the other three drought indices are not reflected in the results section. It is suggested that the calculation results of the SPEI, SSMI, and ETDI be placed in a supplementary file.

**Answer:** Thank you for your suggestion. In fact, the calculation results of the four drought indices (the SPEI, SSMI, ETDI, and GTDI) can be found on the Preprint nhess-2023-41 – Supplement link or on the following link: https://nhess.copernicus.org/preprints/nhess-2024-45/nhess-2024-45-supplement.zip

Line 267-269, the findings from the final month of each season were used to depict drought conditions throughout the season; why not utilize a multi-month average?

**Answer:** Thank you for this comment. Drought indices at different time scales can reflect the dry and wet conditions of the study area at different time periods in the past. This study calculated the drought index at a three-month scale, and the calculation results of each month reflect the drought conditions in the past three months. The drought index in May, August, November and February of each year just reflects the dry and wet conditions of the four seasons of spring, summer, autumn and winter in meteorology.Therefore, we use the drought index in the last month of each season to reflect the dryness and wetness of the season.

In Figure 4, "PCC" can be marked above the legend on the right, and enlarge the names of the two drought indices in each row on the left.

**Answer:** Thank you for your feedback. We have improved Figure 4 as suggested.

[Figure]

Figure 4. Spatial distribution of correlation coefficients in different seasons. The color bar on the right denotes the Pearson's correlation coefficients.

Line 297, "worse" is inappropriately used to describe correlation coefficients (PCC) and should be replaced by "lower."

**Answer:** Thank you for your suggestion. We have replaced "worse" by "lower" here. (Line 305)

Line 314, "their" should be changed to "its".

**Answer:** Thank you. We have followed your suggestion to change "their" to "its" here (Line 322), and we have carefully searched and revised the full text for more similar questions. Thank you again for your careful review.

Line 320-321, "to contrast the weight distribution of SPEI and SSMI in ...", "allocation" may be more suitable than "distribution" here.

**Answer:** Thank you. We have modified this sentence as follows:

Lines 328–330: *"To contrast the weight distribution of SPEI and SSMI in creating the integrated drought indices GTDI and ETDI, the spatial allocation of their weight ratios (SPEI/SSMI) in the WRB is plotted, as shown in Fig. 5."*

Line 328, "comprehensive" should be replaced by "integrated".

**Answer:** Thank you for pointing out this issue. We have replaced "a comprehensive" by "an integrated" as suggested. (Line 336)

Line 343, "as a consequence of comparing GTDI and ETDI, it is discovered that ...", "is" should be changed to "was".

**Answer:** Thank you. According to your suggestion, we have changed this sentence as follows: Line 351: *"as a consequence of comparing GTDI and ETDI, it was discovered that …"*

Line 344-345, "which is essentially congruent with the drought generation mechanism in this basin": what is the drought generation mechanism in Wei River Basin? Please elaborate on this sentence better.

**Answer:** Thank you for this comment. In this study, drought events in the Wei River Basin are dominated by a lack of precipitation. The Standardized Precipitation Evapotranspiration Index (SPEI) is closely related to precipitation. When precipitation is low, the SPEI index will decrease, indicating an increased possibility of a meteorological drought. However, the Standardized Soil Moisture Index (SSMI) is calculated by soil moisture data, reflecting the occurrence of regional drought influenced by the change of soil moisture.

In the construction of GTDI, the weight of the meteorological drought index SPEI is slightly higher than that of the agricultural drought index SSMI, indicating that SPEI, or precipitation, dominates the changes in GTDI more, which is consistent with the occurrence of drought in the Weihe River Basin dominated by precipitation shortage. Therefore, it is mentioned that: Lines 351–354: *"the game theory approach gives an integrated weight geographic distribution compatible with the precipitation-dominated natural drought pattern, which is essentially congruent with the drought generation mechanism in this basin."*

Figures 6 and 7 can be combined into one figure.

**Answer:** Thank you for your feedback. We agree with your suggestion to a certain extent, but considering that it is not easy to arrange the sub-figures after combining the two figures, we still hope to draw both Figure 6 and Figure 7 separately, because it is also more convenient to compare the comprehensive drought index GTDI focused in this manuscript with the other three drought indices.

Line 371, "the soil moisture data used in this study is only 0 to 10cm of soil surface layer", "is" should be changed to "are".

**Answer:** Thank you for your suggestion. We have changed "is" to "are" as suggested. (Line 380)

Figure 9 needs to be streamlined, as the year-month labeling is somewhat repetitive. It is suggested that the three drought indices be marked with only one year-month label under each drought event image.

**Answer:** Thank you for this comment. We have followed your suggestion to modify Figure 9.

[Figure]

Figure 9. Comparison of SPEI, SSMI and GTDI in the spatial evolution of three drought events. The black circle donates the monthly drought centroid.

Line 463-466, "In addition" and "Furthermore" are repeated, "in addition" can be removed.

**Answer:** Thank you for your feedback. We have removed "in addition" as suggested. (Line 473)

**5. Conclusions**

Line 482, add " between "correlation" and "in".

**Answer:** Thank you for this comment. The conclusion section has been streamlined and compressed as suggested by Anonymous Referee #1, and the sentence you mentioned has been deleted.

Line 492, the same as the comment for line 320-321, "allocation" may be more suitable than "distribution" here.

**Answer:** Thank you. We have  modified this sentence as follows:

Lines 495–496: *"This indicates that the GTDI's weight allocation of SPEI and SSMI is more logical and trustworthy."*

Line 511-513, the evolution trend of the GTDI is first presented in the results section, why don't authors summarize the findings of this part in the first conclusion?

**Answer:** Thank you for pointing out this issue. This part of the conclusion is not the key finding of this study. It is just a summary of the evolution trend of drought in the Wei River Basin, based on the evolution trend of GTDI in recent decades and the main findings of this study. In addition, the content you mentioned has been deleted in the revised manuscript as it is not a key finding or key point.

**Review 3**

**Comment on nhess-2024-45, Anonymous Referee #3**

Many thanks for your positive feedback for the results and scientific significance in this study. We greatly appreciate the Reviewer's comments, all suggestions are helpful in improving this manuscript. We are committed to enhancing the quality of our manuscript based on the reviewers' comments. We have carefully studied, considered and responded to all comments point-by-point as follows. For clarity, all comments are given in black and responses are given in blue text.

**Comment 1:** GTDI: a spatially variable weight drought combining two single-type indices SSMI and SPEI for drought hazard causing and bearing impacts changing. Authors claim that GTDI has a greatly high correlation with single-type drought indices (SPEI and SSMI) which is obvious because both indices are used to develop GTDI.

**Answer 1:** Thank you for the positive feedback.We completely agree with the issue that you presented. It should be emphasized the primary advantage of GTDI is its ability to obtain a spatially variable weight distribution pattern between SPEI and SSMI. This allows the integrated drought index GTDI developed by SPEI and SSMI to give a composite state of basin or regional drought hazard causing and bearing conditions. Therefore, to be accurate, the GTDI is relevant and distinct from SPEI and SSMI.

**Comment 2:** This is not an individual validation of GTDI.

**Answer 2:** Thank you for your comment. As everyone is aware, the generation and impact mechanisms of drought are extremely complex, with far-reaching consequences (involving meteorology, hydrology, agricultrue, social economy, etc.).Thus, it is highly difficult to define an absolute "true value" for drought, resulting in a lack of a defined standard or process for evaluating the accuracy of drought indices that reflect the full drought process. SPI, RDI, SMDI, DAI, and EVI are well-known drought indicators, however they only show a portion of the drought's impact. Although the GTDI developed in this manuscript, is described as a integrated drought index, its effectiveness can only be demonstrated by comparative analysis. Considering the drought type represented by the GTDI and the data source used, SPEI, SSMI, and ETDI are empolyed to compare with GTDI in this study to illustrate GTDI's reasonableness and effectiveness .

**Comment 3:** Line 28: "GTDI exhibits the gaming feature" What are the features? The major question is how the author came up with this equation.

**Answer 3:** Thank you for pointing out this issue. The gaming feature of game theory is that it can try to find an optimal allocation method that maximizes the interests of each participant through mathematical analysis [24]. Any change in the participants will cause a corresponding change in the game situation, as we stated in lines 174 to 183 in the revised manuscript.

As for the equation you mentioned, we only applied this method to the development of an integrated drought index based on the principles of game theory, and the original equation and the revelant description can be found in the listed references [24,25].

[24] Jato-Espino, D. and Ruiz-Puente, C.: Bringing Facilitated Industrial Symbiosis and Game Theory together to strengthen waste exchange in industrial parks, Sci. Total Environ., 771, 145400, https://doi.org/10.1016/j.scitotenv.2021.145400, 2021.

[25] Lai, C., Chen, X., Chen, X., Chen, X., Wang, Z., Wu, X., and Zhao, S.: A fuzzy comprehensive evaluation model for flood risk based on the combination weight of game theory, Nat. Hazards., 77, 1243-1259, https://doi.org/10.1007/s11069-015-1645-6, 2015.

**Comment 4:** Are there any sound criteria that support this form?

**Answer 4:** Thank you for your comment. Sorry, we are not sure which part of the manuscript you are referring to. After careful inspection, we think you should be referring to Table 2.

In response to your concerns, we have added a reference to Table 2 to explain the basis for the drought classification criterion we used as follows:

Line 196: "*Table 2. Drought classification criteria for the SPEI, SSMI, GTDI and ETDI [26].*"

[26] Huang, F., Liu, L., Gao, J., Yin, Z., Zhang, Y., Jiang, Y., and Fang, W. (2023). Effects of extreme drought events on vegetation activity from the perspectives of meteorological and soil droughts in southwestern China. Science of The Total Environment, 903, 166562.

**Comment 5:** The soil moisture dataset resolution is very high for a very course and a very small catchment means all the regions could exhibit similar values how do authors distinguish?

**Answer 5:** Thank you for bringing up these issues. Sorry, we don't quite understand what you would like to express. After careful consideration, we conjecture that your concern is that the spatial resolution of the soil moisture dataset is "very low" rather than "very high." In response to your doubts, we would like to say that your concerns are indeed a difficult problem in our work. Because the accuracy of the soil moisture datasets currently available is indeed limited, we can only use relatively reliable datasets for this study. This could cause soil moisture in a very small catchment to exhibit a similar value. Once a soil moisture dataset with higher spatial resolution and reliable accuracy is released, we will consider using more detailed data for future studies.

**Comment 6:** What is the basis for classifying the GTDI? Line 194 "calculating approach of SSMI in this study is comparable to that of SPEI, while GTDI and ETDI are built on SSMI and SPEI" GTDI is using a weighted approach, and it may reflect different drought intensity/severity. Thus, the classification approach could not be straightforward.

**Answer 6:** Thank you for this comment. It should be explained that the classification of GTDI learns from the categorization of the meteorological drought index (SPEI). As indicated in our response to your comment 4, we have added a reference to explain this classification.

In fact, calculating GTDI using the game theory weight allocation method does not drastically affect drought intensity or severity, but it is a beneficial combination of SPEI and SSMI. Furthermore, our correlation research revealed that GTDI has a very strong association with single-type drought indices (the SPEI and SSMI), indicating that there is reliable consistency between them. As a result, it appears more reasonable to apply a consistent drought categorization for them, whereas classification reasonability is difficult to verify if using another classification method for GTDI, which may even lead to greater confusion and inappropriateness.

**Comment 7:** How did authors build ETDI? Section 3.2 only shows the GTDI process.

**Answer 7:** Thank you for this comment. We apologize for the inadequate presentation of key methods.The detailed calculation process of the ETDI index has been stated in the supplementary material as suggested by Anonymous Referee #2, which can be found in Supplement S3.

**Comment 8:** Temporal evaluation of GTDI is needed to present along SPEI and SSMI.

**Answer 8:** Thank you for your positive feedback. In fact, we have put GTDI together with SPEI and SSMI for temporal evaluation and analysis, as presented in Figure 8. Figure 8 shows the comparison of drought identification trajectories of the GTDI, SPEI, and SSMI over time, and based on this, we analyzed the differences and connections between the integrated drought index GTDI and the single drought indices SPEI and SSMI.

**Comment 9:** Figure 3: The results are not meaningful. What scale is used for calculating drought with GTDI? Is this drought tendency mild, moderate, or extreme?

**Answer 9:** Thank you for this comment. It should be explained that the monthly-GTDI is calculated based on SPEI-3 and SSMI-3.

We apologize for not fully understanding your concerns regarding the question, "*Is this drought tendency mild, moderate, or extreme?*".

The drought tendency in the Wei River Basin is aggravating, as we stated in lines 271 to 272 of the manuscript. (lines 279 to 280 in the revised manuscript)

"*Therein, the linear tendency rate of GTDI is -0.024/10a, illustrating that the drought in the WRB is aggravating.*"

After careful consideration of your doubts, we speculate that you may would like to know the drought grade corresponding to the monthly GTDI index. Therefore, we have improved Figure 3 (a) as follows:

[Figure]

Figure 3. Temporal evolution characteristics of integrated drought in the Wei River Basin from 1950 to 2020 (a), and spatial distribution of drought trends in different seasons (b-e). The symbol "**" donates the change is significant, and the percentage means the area proportion of different trend types.

**Comment 10:** GTDI is developed to present an agricultural drought. Right? Individual verification is needed.

**Answer 10:** Thank you for this comment. It is necessary to clarify that the GTDI isn't developed to present an agricultural drought, it is an integrated drought index by combining two single-type indices: the meteorological drought index (SPEI) and the agricultural drought index (SSMI).

**Comment 11:** The efficacy verification in Table 6 only presents some percentage numbers. Why choose these specific months, which period? GTDI is overestimating the drought ratio because this river basin has a very small area where seasonal drought happens. Moreover, Fig. 6 and 7 present what is beyond my understanding. What is the purpose of showing satellite image?

**Answer 11:** Thank you for raising these issues. March to August was selected as the validation period in Table 6 because the surface vegetation during this period is more sensitive to changes in soil moisture. When drought occurs, vegetation growth will be restricted, and the LAI will be lower than the value of the drought-free month in the same period. Therefore, in this study, we used the change in the mean values of LAI in drought and non-drought months to verify whether the drought identified by GTDI, SPEI, SSMI, and ETDI is effective and reliable, that is, comparing the LAI mean values of drought and non-drought months identified by the drought index in the same month over many years. If the LAI mean

values of drought months is lower than the LAI means value of non-drought months, it illustrates that the drought identified by the drought index is effective and reliable, as we stated in lines 207 to 237 of Section 3.4.2 of the manuscript.

The data listed in Table 6 are not the drought ratio for each month, but the efficacy ratio of drought index in identifying drought, as the name of the table suggests: "*Table 6. The efficacy ratios of four drought indices in different validation months*"

Figure 6 shows the spatial distribution of GTDI's efficacy in identifying drought in the Wei River Basin, while Figure 7 shows the spatial distribution of SPEI, SSMI, and ETDI's efficacy in identifying drought in the Wei River Basin, respectively. The legends "Fine" and "Poor" in the figures indicate the drought recognition performance of the drought index, that is, "Fine" means that the drought index accurately monitored the occurrence of drought, while "Poor" means that the drought index did not monitor the occurrence of drought, as we stated in line 237 of Section 3.4.2 of the manuscript (lines 238 to 239 in the revised manuscript).

The satellite image shown in Figure 6 is to illustrate the reason behind the poor and concentrated drought recognition performance of GTDI. As shown in Figure 6, the grid points with poor performance in June and August are concentrated in the forest area, which is the dark green area in the WRB's northeast hinterland. Forests have more access to deeper soil moisture than farming land and grassland [27,28], resulting in forests having higher drought tolerance than other terrestrial vegetation types [29,30]. The soil moisture data used in this study is only 0 to 10cm of soil surface layer, which could explain why GTDI's drought diagnosis ability in the forest region is skewed, as we stated in lines 365 to 372 of the manuscript (lines 374 to 381 in the revised manuscript).

[Figure]

Figure 6. The spatial distribution of GTDI's efficacy in identifying drought in the Wei River Basin. Subfigures (a)-(f) depict the findings from March to August, and (g) displays a satellite image of the Wei River Basin. "Fine" means that the drought index accurately captured the occurrence of drought, while "Poor" means that the drought index did not capture the occurrence of drought.

[27] Xu, H., Wang, X., Zhao, C., and Yang, X. Diverse responses of vegetation growth to meteorological drought across climate zones and land biomes in northern China from 1981 to 2014, Agric. For. Meteorol., 262, 1-13, https://doi.org/10.1016/j.agrformet.2018.06.027, 2018.

[28] Bai, Y., Liu, M., Guo, Q., Wu, G., Wang, W., and Li, S. Diverse responses of gross primary production and leaf area index to drought on the Mongolian Plateau, Sci. Total Environ., 902, 166507, https://doi.org/10.1016/j.scitotenv.2023.166507, 2023.

[29] Jiang, W., Wang, L., Feng, L., Zhang, M., and Yao, R. Drought characteristics and its impact on changes in surface vegetation from 1981 to 2015 in the Yangtze River Basin, China, Int. J. Climatol., 40, 3380-3397, https://doi.org/10.1002/joc.6403, 2020.

[30] Chen, Q., Timmermans, J., Wen, W., and van Bodegom, P.M. A multi-metric assessment of drought vulnerability across different vegetation types using high resolution remote sensing, Sci. Total Environ., 832, 154970, https://doi.org/10.1016/j.scitotenv.2022.154970, 2022.

**Comment 12:** What do you mean fine, poor?

**Answer 12:** Thank you very much for your comment. The legends "Fine" and "Poor" in Figures 6 and 7 indicate the drought recognition performance of the drought index, that is, "Fine" means that the drought index accurately monitored the occurrence of drought, while "Poor" means that the drought index did not capture the occurrence of drought, as we stated in line 237 of Section 3.4.2 of the manuscript. In response to your doubts, we have added a brief explanation to explain this legend annotation. (lines 367 to 368 in the revised manuscript).

**Comment 13:** Fig. 9: it could be seen that drought is moderate in this basin, Thus authors are advised to expand the region for proper verification of GTDI.

**Answer 13:** Thank you very much for your comment. Figure 9 is used to compare the spatial evolution of the SPEI, SSMI, and GTDI during the three drought events rather than describe the drought level in this basin. The three drought events we selected are to facilitate the comparison of the spatial evolution of droughts identified by the SPEI, SSMI, and GTDI, as well as explore their connection and development discipline.

Sorry, but we don't understand why you perceive the drought in the basin to be moderate and need to expand the area to properly verify GTDI. We are wondering if you would be willing to explain it in detail for us so that we can better understand and solve it. Among the three drought events shown in Figure 9, the 2000 drought event was clearly a severe drought event throughout the whole basin, because in May 2000, drought occurred in the entire Weihe River basin, and the average drought intensity was greater than 1, as listed in Table 7.

**Technical corrections**

**Editor Shreedhar Maskey:**

**Comment:** In the Abstract, Ln 17 and 22, I suggest to replace the word 'induced' by 'introduced'. No other comments.

Answer : Thank you very much for your suggestion. We have replaced the word 'induced' by 'introduced' in lines 17 and 22.

**Executive editor Kai Schröter:**

**Comment1:** Fig. 1 to include also the data sources with references for elevation, temperature and precipitation shown in the maps.

Answer1 : Thank you very much for your comment. We have added corresponding references for elevation, temperature and precipitation data shown in Fig. 1, as follows:

[Figure]

Figure 1. A map of the Wei River Basin. Subfigures (a) shows the geographical location of the Wei River Basin in China, (b) displays the spatial distribution of elevation **(Zhang, 2021)** in the Wei River Basin, (c) and (d) demonstrate the annual precipitation and temperature **(Peng et al., 2019)** of the WRB.

[31] Zhang, Y.: 30m resolution digital elevation model (DEM) data of Weihe River Basin, National Cryosphere Desert Data Center, https://www.doi.org/10.12072/ncdc.WRiver.db0009.2021, 2021.

[32] Peng, S., Ding, Y., Liu, W., and Li, Z.: 1 km monthly temperature and precipitation dataset for China from 1901 to 2017, Earth Syst. Sci. Data, 11, 1931–1946, https://doi.org/10.5194/essd-11-1931-2019, 2019.

**Comment2:** Fig. 7 Please add meaning of 'fine' and 'poor' to the caption, as for the caption of Fig. 6.

Answer2 : Thank you very much for your suggestion. We have added meaning of 'fine' and 'poor' to the caption for Fig. 7 as for the caption of Fig. 6, as follows:

[Figure]

March    April    May    June    July    August

**Legend**    Fine    Poor

Lines 373-375: *"**Figure 7.** The spatial distribution of efficacy in identifying drought of the ETDI, SPEI and SSMI. **"Fine" means that the drought index accurately captured the occurrence of drought, while "Poor" means that the drought index did not capture the occurrence of drought.***"

Finally, we sincerely appreciate the positive feedback and the valuable insights proposed by peer reviews. Your comments and concerns are valuable and meaningful to improving the manuscript. At the same time, we want to express our gratitude to the editors for their time in processing this manuscript. Thank you very much.